# Human click-based echolocation: Effects of blindness and age, and real-life implications in a 10-week training program

**Liam J. Norman‡, Caitlin Dodsworth, Denise Foresteire, Lore Thaler‡ \***

Department of Psychology, Durham University, Durham, United Kingdom

‡ These authors share first authorship on this work.
* lore.thaler@durham.ac.uk

**Data Availability Statement:** All relevant data are within the manuscript and its Supporting information files.

## Abstract

Understanding the factors that determine if a person can successfully learn a novel sensory skill is essential for understanding how the brain adapts to change, and for providing rehabilitative support for people with sensory loss. We report a training study investigating the effects of blindness and age on the learning of a complex auditory skill: click-based echolocation. Blind and sighted participants of various ages (21–79 yrs; median blind: 45 yrs; median sighted: 26 yrs) trained in 20 sessions over the course of 10 weeks in various practical and virtual navigation tasks. Blind participants also took part in a 3-month follow up survey assessing the effects of the training on their daily life. We found that both sighted and blind people improved considerably on all measures, and in some cases performed comparatively to expert echolocators at the end of training. Somewhat surprisingly, sighted people performed better than those who were blind in some cases, although our analyses suggest that this might be better explained by the younger age (or superior binaural hearing) of the sighted group. Importantly, however, neither age nor blindness was a limiting factor in participants' rate of learning (i.e. their difference in performance from the first to the final session) or in their ability to apply their echolocation skills to novel, untrained tasks. Furthermore, in the follow up survey, all participants who were blind reported improved mobility, and 83% reported better independence and wellbeing. Overall, our results suggest that the ability to learn click-based echolocation is not strongly limited by age or level of vision. This has positive implications for the rehabilitation of people with vision loss or in the early stages of progressive vision loss.

## 1. Introduction

There is a substantial body of research investigating how the brain adapts in the context of visual sensory deprivation (for reviews see [1–8]). With respect to behavioural performance in spatial hearing tasks, it has been found that people who are blind show improved localization of sounds in the periphery [9], improved spatial tuning in the periphery [10], improved monoaural localization [11], and improved discrimination of distances of sound sources [9,12]. On

**Funding:** The work was funded by a grant of the Biotechnology and Biological Sciences Research Council United Kingdom (www.bbsrc.ukri.org), awarded to LT (award number: BB/M007847/1) and a grant from the Network for Social Change (www.thenetworkforsocialchange.org.uk/), awarded to LT (award number: 30/7490). The funders had no role in study design, data collection and analysis, decision to publish, or preparation of the manuscript.

**Competing interests:** The authors have declared that no competing interests exist.

the other hand, people who have been blind from an early age may struggle more when localizing the vertical location of sounds [13], or judging relative spatial position between three successive sounds [14]. In sum, visual deprivation is associated with complex neuroplastic changes with respect to spatial hearing performance.

Echolocation is a particular spatial hearing skill, namely the ability to use reflected sound to get information about the environment. Even though echolocation is primarily associated with bats, it is by now well established that humans are able to use it as well [15–17]. A distinction can be made between passive and active echolocation. For passive echolocation, one listens to emissions and echoes where emissions have been made by sources other than oneself, e.g., ambient sound fields, or another person speaking, making mouth-clicks etc. For active echolocation, one makes their own emissions and uses echoes arising from those, e.g., echoes from one's own mouth clicks, footsteps, cane taps, etc. Laboratory research has shown that click-based echolocation provides sensory advantages above and beyond passive echolocation via ambient sound fields, e.g. [18–20] or active echolocation using footsteps or cane-taps [21,22].

Here we used click-based echolocation to investigate if blindness and age are relevant factors for acquiring a complex spatial hearing skill. We chose click-based echolocation for this purpose, because people rarely use this skill, and it can therefore provide a good baseline from which to start. It is an open question if blindness *per se* may put people at an advantage for click-based echolocation (for review [15]) or if instead experience with this skill is most important [22–27]. There have been studies investigating how sighted people learn click-based echolocation across multiple sessions [18,19,28,29]. Yet, a training study involving people who are blind is still missing, and it is an open question if blindness is a relevant factor for acquiring a complex spatial hearing skill such as echolocation.

From a basic science perspective, investigating how people learn to echolocate and how this is related to vision loss and other aspects of their hearing would provide information on how the brain adapts in response to sensory deprivation and to learning a new skill. This more general question was addressed in one previous training study in which sighted and blind people used a visual-to-tactile sensory substitution device. It was found that after 4 sessions, blind people performed better than sighted controls [30], suggesting that visual deprivation may put people at an advantage for learning a new sensory skill. Importantly, if this is a general principle, we might expect that it would also apply to learning a complex auditory skill such as click-based echolocation, so that people who are blind would learn better than those who are sighted.

From an applied science perspective, click-based echolocation may offer behavioural benefits for people who are blind, but previous reports on this have been limited to a correlational approach [31]. Thus, knowing more about learning this skill and knowing more about how it might transfer to everyday life in people who are blind is important for instruction. Furthermore, in an increasingly ageing population, age-related vision loss affects more people now than ever before and, in fact, about 80% of people with vision loss are 50 years or older [32]. Whilst the older human brain might nonetheless adapt to new challenges, the neuroplastic processes involved may differ between older and younger people [33,34]. Yet, to date there has only been limited work investigating the learning of novel sensory skills in older age groups. Specifically, there is some evidence that, when learning to navigate a virtual environment using a sensory substitution device, older people do not learn as well as younger people do [35], suggesting merit of further investigation of the effect of age in research on neuroplasticity.

The current study investigated how people learn to echolocate over the course of 10 weeks (20 sessions, each between 2 and 3 hours in length). People were trained in three different tasks (size discrimination, orientation perception, virtual navigation) and also navigated using

echolocation in natural environments for 30 minutes each session. Participants were either sighted (n = 14) or blind (n = 12), and of various ages (sighted: min 21 yrs, max 71 yrs, median 26 yrs; blind: min 27 yrs, max 79 yrs, median 45). To validate our paradigms and to benchmark our tasks we also tested seven blind experts in click-based echolocation. We also measured audiometric thresholds and sensitivities to level and spectral changes. For people who were blind we also used a follow-up survey 3 months after the training, to determine how the training had affected their mobility, independence and wellbeing. As part of the training, all participants also performed a control task that did not require click-based echolocation.

Part of the data (sighted participants' performance in the virtual training task) has been reported previously [28].

## 2. Methods

### 2.1 Ethics statement

All Procedures followed the British Psychological Society code of practice and the World Medical Association's Declaration of Helsinki. The experiment had received ethical approval by the Ethics Advisory Sub-Committee in the Department of Psychology at Durham University (Ref 14/13). All participants gave written informed consent to take part in this study. Participants who were sighted and participants who were blind received £6/hr and £10/hr, respectively, to compensate them for their effort and time taking part.

### 2.2. Participants

Fourteen sighted participants (SCs; 8 males, 6 females) took part (ages: 21, 21, 22, 22, 23, 24, 25, 27, 32, 35, 38, 48, 60, and 71; mean = 33.5, SD = 15.8, median = 26). All reported to have normal or corrected to normal vision and no prior echolocation experience (based on self-report). Twelve blind participants (BCs; 6 males, 6 females) with no prior experience in click-based echolocation took part (details shown in Table 1). In our sample, all BCs had cause of vision loss present from birth. All were diagnosed as legally blind in childhood, with only two official diagnoses at an age that might have coincided with onset of puberty, or may have been after onset of puberty (i.e. 13 yrs and 10 yrs; BC6 and BC2), but again with vision impairment having been present from birth. Thus, the majority of our participants are classified as early blind. All our blind participants were independent travellers and all had received mobility and orientation training as part of visual impairment (VI) habilitation and VI rehabilitation that is provided to people with VI in the UK. The group of blind participants was significantly older (mean: 48.3, SD: 15.4, median = 45) than the group of sighted participants (t(24) = 2.413, p = .024).

To validate and benchmark our tasks we also tested blind echolocation experts (EEs) in single sessions of each task (i.e. no training took place). Our requirements for classing an individual as an echolocation expert were that they reported using click-based echolocation on a daily basis for more than 10 years. In our sample, five out of seven echolocation experts (EEs) had cause of vision loss present from birth and were diagnosed as legally blind from birth/within the first year of life. The remaining two experts received official diagnoses at an age that might have coincided with onset of puberty, or may have been after onset of puberty (i.e. 14yrs and 12 yrs; EE3 and EE7). Thus, the majority of our echolocation expert participants are classified as early blind. Not every echolocation expert took part in each task; details of each echo expert and the tasks they took part in are listed in Table 2. With the exception of one blind participant (BC8, aged 72 yrs) who wore hearing aids to compensate for age related hearing loss, all participants had normal hearing appropriate for their age group (ISO 7029:2017) assessed using pure tone audiometry. For purposes of testing, the participant with hearing aids did not wear

**Table 1. Details of participants who were blind.**

| ID | Gender | Age | Degree of vision loss | Cause and age at onset of vision loss | Echolocation use prior to taking part |
|---|---|---|---|---|---|
| BC1 | F | 60 | Total blindness in left eye; some peripheral vision in right eye. | Stichler's syndrome. Retinal sciasis, from birth with increasing severity | Some experience; very little regular use |
| BC2 | M | 54 | Residual bright light perception | Retinitis pigmentosa. Official diagnosis age 10 yrs. Gradual sight loss from birth | Some experience; very little regular use |
| BC3 | M | 39 | Residual bright light perception | Retinitis pigmentosa. Gradual sight loss from birth. Official diagnosis in early childhood (no exact age remembered but was known when commencing school, i.e. age 5yrs). | None |
| BC4 | M | 46 | Total blindness | Ocular albinism. Gradual sight loss from birth. | Some experience; very little regular use |
| BC5 | F | 36 | Bright Light detection | Unknown cause; from birth. | None |
| BC6 | M | 37 | Tunnel vision (<5 deg) and decreased acuity (< 20/200) in both eyes. | Retinitis pigmentosa. Gradual sight loss from birth. Official diagnosis age 13yrs. | None |
| BC7 | M | 48 | Total blindness in left eye; residual bright light perception in right eye | Severe childhood glaucoma; 3 months old | None |
| BC8 | F | 72 | Bright Light detection | Retinitis Pigmentosa. Gradual sight loss from birth. Official diagnosis in early childhood (no exact age remembered but was known when commencing school, i.e. age 5yrs). | None |
| BC9 | F | 79 | Some blurred foveal vision; prone to bleaching | Rod Cone Dystrophy. Birth | None |
| BC10 | F | 44 | Total Blindness right eye; bright light detection left eye | Microphtalmia and Glaucoma; right eye enucleated aged 39yrs | None |
| BC11 | F | 27 | Left eye ca. 1 deg of foveal vision left with reduced acuity (<20/200); right eye bright light detection | Leber's Amaurosis and Cataracts. Birth. | None |
| BC12 | M | 38 | Tunnel vision (<2 deg) and decreased acuity (< 20/200) in both eyes. | Retinitis Pigmentosa and other retinal pathology (unknown). Official diagnosis in early childhood (no exact age remembered but was known when commencing school, i.e. age 5yrs). | None |

Unless otherwise stated, official diagnosis from birth/within first year of life.

**Table 2. Details of blind echolocation experts.**

| ID | Gender | Age | Degree of vision loss | Cause and age at onset of vision loss | Echolocation use | Size Task | Orientation Task | Virtual Navigation Task |
|---|---|---|---|---|---|---|---|---|
| EE1 | M | 49, 50 | Total blindness | Enucleation due to retinoblastoma at 13 months | Daily; since early childhood/ no exact age remembered | X 49 | X 49 | X 50 |
| EE2 | M | 31, 32, 35 | Bright light detection | Gradual sight loss since birth due to glaucoma. | Daily; since 12 yrs | X 31 | X 32 | X 35 |
| EE3 | M | 33 | Total blindness | Optic nerve atrophy at 14 yrs; official diagnosis age 14 yrs | Daily since 15yrs | X | X | |
| EE4 | F | 41 & 43 | Total blindness | Leber's congenital amaurosis; birth | Daily since 31yrs | | X 41 yrs | |
| EE5 | M | 19 | Total blindness | Leber's congenital amaurosis; start at birth total blindness at 3yrs | Daily; since early childhood/ no exact age remembered | | X | |
| EE6 | M | 49 | Total blindness | Retinoblastoma; enucleated at 18 and 30 months | Daily use since 8yrs | | | x |
| EE7 | M | 24 | Total blindness | Sudden loss at 12yrs due to unknown causes; enucleation at 19yrs; official diagnosis at 12 yrs. | Daily since 12yrs | | | X |

Testing took part across years for some participants. If ages differed for different experiments, ages are indicated for each experiment in the last columns. Unless otherwise stated, official diagnosis from birth / within first year of life.

their aids during any of the experimental testing sessions. All participants who had any residual vision were tested under blindfold.

## 2.3. Apparatus and procedures

All testing (except navigation in natural environments) took place in a sound-insulated and echo-acoustic dampened room (approx. 2.9 m × 4.2 m x 4.9 m) lined with foam wedges (cut-off frequency 315 Hz) in the department of psychology at Durham University.

All computer-based tests were run with MATLAB R2018b (The Mathworks, Natick, MA) and modified functions from the Psychtoolbox library [36] on a laptop (Dell Latitude E7470; Intel Core i56300U CPU 2.40; 8GB RAM; 64-bit Windows 7 Enterprise) with external sound card (Creative Sound Blaster External Sound Card Model SB1240; Creative Technology Ltd., Creative Labs Ireland, Dublin, Ireland; 24 bit and 96 kHz). Stimuli were presented through headphones (Etymotic ER4B; Etymotic Research, Illinois, USA).

Participants were introduced to the tasks and mouth clicks prior to the first session. Training took part over 20 sessions on 20 separate days spread over a 10-week period. During each training session participants performed a size discrimination task, an orientation perception task, and a virtual navigation task—all inside the lab. Either in the beginning or at the end of the session they also used mouth clicks to navigate outside the lab in natural indoor or outdoor settings for 30 minutes. Tasks were done in different orders across participants and sessions. For practical reasons, the natural navigation task was always done either in the beginning or the end of a session (in approximately equal parts). The other, lab-based, tasks were run in different orders across participants, with participants being able to choose the order if they wished. The rationale for this approach was that we wanted to avoid order effects, whilst also working in a participant-led way by maintaining participants motivation across a demanding training schedule. All tasks were always done under supervision of an experimenter. Three of the authors (LN, LT and CD) led instruction. To ensure consistency, the principal investigator (LT) gave in house training to all instructors prior to any training commencing, and (determined by availability of the instructors) at least two instructors were present for any session. Instructors were not assigned in any specific fashion to participants or sessions, but this was determined by availability. In addition to the echolocation training sessions there were two sessions during which we performed a number of basic hearing tests. Any session lasted between 2 and 3 hours, depending on how many and how long breaks participants wished to take. Fig 1 shows an illustration of the general procedure for data collection for a single participant.

**2.3.1. Demographic measurements of hearing ability.** *2.3.1.1. Audiometry.* Pure tone audiometry (0.25, .5, 1, 2, 4, 8 kHz) was performed using an audiometer (Interacoustics AD629, Interacoustics, Denmark) and the Hughson Westlake procedure. For one participant who was sighted (aged 22 yrs) an error occurred and data were not saved.

Following the analysis by [37] we used each participant's audiometry values to calculate their average hearing sensitivity, i.e. we averaged their threshold values across tested frequencies. In addition, following the analysis of [37] we also calculated the average absolute binaural difference in hearing level across frequencies, i.e. for each frequency separately we calculated the absolute difference between right and left ears and these were subsequently averaged across frequencies. For the participant using a hearing aid (but tested without hearing aid in our experiments), threshold at 8 kHz (right ear) could not be determined and we used only frequencies up to 4 kHz to calculate these measures.

Average hearing level did not differ significantly between BCs (mean: 13.33, SD: 14.53) and SCs (mean: 5.19; SD: 10.88; $F(1,23) = 2.539$; $p = .125$; $\eta^2_p$: .099). Average absolute binaural

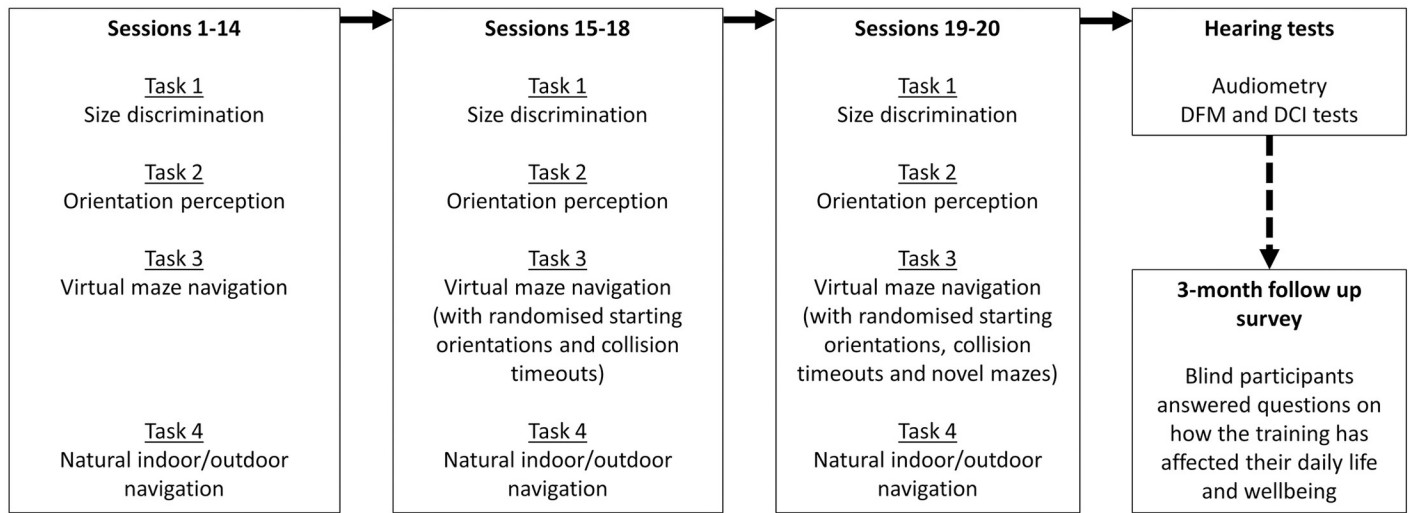

**Fig 1. An illustration of the general procedure for data collection for a single participant.** Note that tasks were done in different orders across participants and sessions, and this flowchart is just for illustration. In each of the 20 training sessions, each participant performed four separate tasks—size discrimination, orientation perception, virtual maze navigation, and real indoor/outdoor navigation. Please see the main text for descriptions of each of the tasks. For practical reasons, the natural navigation task was always done either in the beginning or the end of a session (in approximately equal parts). The other, lab-based, tasks were run in different orders across participants, with participants being able to choose the order if they wished. An additional session was used to acquire basic measurements of participants' hearing (audiometry and DFM/DCI hearing tests)–please see the main text for details. Note that for some participants these were acquired before training began, rather than after (as indicated in the figure). For blind participants, they were contacted 3 months following training completion to complete a survey.

differences in hearing level also did not differ significantly between BCs (mean: 6.46, SD: 3.26) and SCs (mean: 5.38; SD: 2.69; $F_{(1,23)}$ = .812; p = .377; $\eta^2_p$: .034). Older age was associated with higher thresholds ($r_{(N=25)}$ = .708, p < .001), which is expected, and also average absolute binaural difference ($r_{(N=25)}$ = .575; p = .003).

*2.3.1.2. Detection of Sound Frequency Modulation (DFM).* This test has been used previously to determine if an individual's ability to determine changes in spectral frequency sound may be related to their ability to echolocate, e.g., [20,38,39]. The test measures participants' ability to detect a change in the frequency (pitch) of a tone. See S1 File for further details.

We analyzed DFM values at 2000 Hz and 500 Hz test frequencies separately. Overall, performance was in line with values reported elsewhere [38,39], and did not differ between BCs and SCs. Specifically, there was no group difference in DFM values for either 2000 Hz (BCs mean: .438, SD: .233; SCs mean: .541, SD: .151; $F_{(1,23)}$ = 1.236; p = .278; $\eta^2_p$: .051) or 500 Hz (BCs mean: .188, SD: .222; SCs mean: .322, SD: .238; $F_{(1,23)}$ = .2.101; p = .161; $\eta^2_p$: .084), and DFM was not correlated with age ($DFM_{2000}$:$r_{(N=25)}$ = -.190; p: .368; $DFM_{500}$: $r_{(N=25)}$ = -.320; p = .120).

*2.3.1.3. Detection of Changes in Sound Intensity (DCI).* This test has been used previously to determine if an individual's ability to determine changes in sound intensity may be related to their ability to echolocate, e.g., [20,38,39]. The test measures participants' ability to detect a change in intensity (loudness) of a tone. See also S1 File for further details.

We analyzed DCI values at 2000 Hz and 500 Hz test frequencies separately. Overall, performance was in line with values reported elsewhere [38,39], and did not differ between BCs and SCs. Specifically, there was no group differences in DCI values for either 2000 Hz (BCs mean: .269, SD: .255; SCs mean: .416., SD: .241; $F_{(1,23)}$ = 2.211; p = .151; $\eta^2_p$: .088) or 500 Hz (BCs mean: .577, SD: .173; SCs mean: .561, SD: .316; $F_{(1,23)}$ = .024; p = .879; $\eta^2_p$: .001), and DCI was not correlated with age ($DCI_{2000}$: $r_{(N=25)}$ = .029; p: .890; $DCI_{500}$: $r_{(N=25)}$ = .305; p = .138).

Side view

Front view

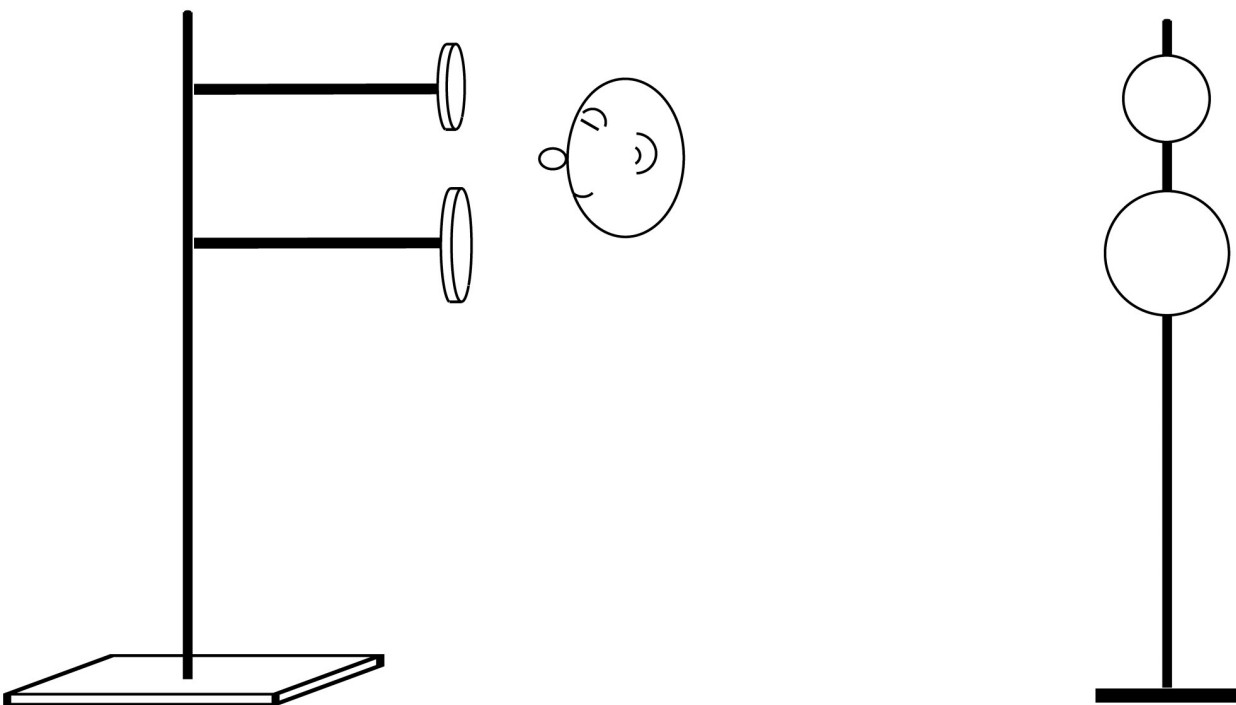

**Fig 2. Illustration of the apparatus used for the size discrimination task.** The (larger) reference disk was used in every trial, and placed either on the top or the bottom bar (here: on the bottom). One of the five (smaller) comparison disks was placed on the remaining, free bar.

**2.3.2. Training tasks.** *2.2.2.1. Size Discrimination Task and Control Task (no click).* This task was introduced by [19] and has been shown to be sensitive to differences in blind peoples' ability to use click-based echolocation based on long-term experience, and to be sensitive to improvement in performance through training in people who are sighted, e.g. [18,19].

**Apparatus**: The apparatus consisted of a vertical round metal pole (1 cm circular diameter), from which two horizontal round metal poles (1 cm diameter) protruded 50 cm towards the participant. The horizontal crossbars were used to mount circular discs made from 0.3 cm foam board. The discs were mounted with a small pin on their back. The largest disc (the reference disc) was 25.4 cm in diameter. The five comparison discs had diameters of 5.1 cm, 9 cm, 13.5 cm, 17.5 cm and 22.9 cm. Fig 2 illustrates the setup.

**Task and Procedure**: Participants' task was to determine if the larger (reference) disk had been located on the top or bottom bar and to signal their response with a silent hand signal. Participants stood in front of the apparatus facing the discs and were encouraged to move their head, with the restriction that they should remain at the same distance (i.e. only up, down, left, right movement was allowed). The height of the apparatus was adjusted so that their ear was equidistant to the top and bottom horizontal bar of the apparatus. The experimenter monitored the participant's distance from the apparatus at all times. There were 30 trials in total, during which each comparison disk was used 6 times (three times on the top and three times on the bottom bar) in random order. Each trial followed the same sequence. First, participants blocked their ears with their index fingers whilst the experimenter positioned the

two disks on the horizontal bars. The (larger) reference disk was used in every trial, and placed either on the top or the bottom bar (15 trials each). One of the five (smaller) comparison disks was placed on the remaining, free bar. Once placement had been completed, the experimenter stepped behind the participant and tapped their shoulder to signal they should unblock their ears. First, participants had 14 seconds to make a guess as to where the larger (reference disk) had been located. This judgment required no clicking and thus served as a control task for other training tasks. A previous study [19] originally introduced this no-click task to control for acoustic conditions like ambient noise, but here we also used it to determine training effects that are not specific to click-based echolocation. For example, if participant's passive echolocation skills were to improve in some way, or if they were to develop a strategy to determine the location of the larger disk based on cues not related to click-echoes, they would improve on this task. Thus, any change in performance in the no-click task would indicate benefits of the training that are not specific to improvement in click-based echolocation. After the participant had given their no-click response, they then started making mouth clicks for a maximum allowed time of 14 seconds. If still making clicks at 14 seconds, a further shoulder tap was given to prompt a judgement. Feedback about where the larger (reference disk) had been located was given verbally to participants after each trial. Participants were initially placed 33 cm from the disks. When a participant performed at 90% or better in two successive sessions they were moved 33 cm further away, and this rule was applied to any subsequent distance (i.e. 66 cm, 99 cm, etc). These distances and strategy had been chosen based on piloting where people new to echolocation found the task easier at closer distances, and so that the task would remain engaging for people as they improved. Echolocation experts performed their single session at 100 cm distance. This distance was chosen because experts said that they preferred this distance over closer or farther distances, i.e. they felt that the task was easiest for them at this distance.

*2.3.2.2. Orientation perception task*. People with expertise in click-based echolocation can use echoes to determine the contour shape of sound-reflecting surfaces, including the orientation (vertical, horizontal) of an elongated rectangle, whilst people who are new to echolocation struggle with this [23]. The orientation perception task was designed to determine how people improve in this skill through training.

**Apparatus**: The apparatus consisted of a vertical round metal pole (0.5 cm circular diameter), onto which a 80-cm x 20-cm rectangle made from 0.3-cm foam board was mounted with a pin at the back of the board. The rectangle could be rotated so to orientations of either vertical, right side up (45°), horizontal, or left side up (135°), always facing the participant. Fig 3 illustrates the setup.

**Task and Procedure**: Participants' task was to use mouth clicks to determine if the rectangle was oriented vertically, horizontally, right side up or left side up and to signal their response with a silent hand signal. The general procedure for this task (including setting stimulus distance etc.) followed that of the size discrimination task. Participants stood in front of the apparatus facing the rectangle, with the height of the apparatus adjusted so that their mouth was located at the centre of the rectangle. There were a total of 24 trials, during which each orientation was used 6 times in random order. Participants had 20 seconds to make mouth clicks on each trial.

*2.3.2.3. Virtual navigation task*. This task was introduced by [28] and has been shown to be sensitive to differences in blind peoples' ability to use click-based echolocation based on long-term experience, and to be sensitive to improvement in performance through training in people who are sighted. The task was computer-based, as this allowed us to test a larger number of trials and spatial configurations in a shorter period of time as compared to if one had to construct these spaces physically.

## Side view

## Front view

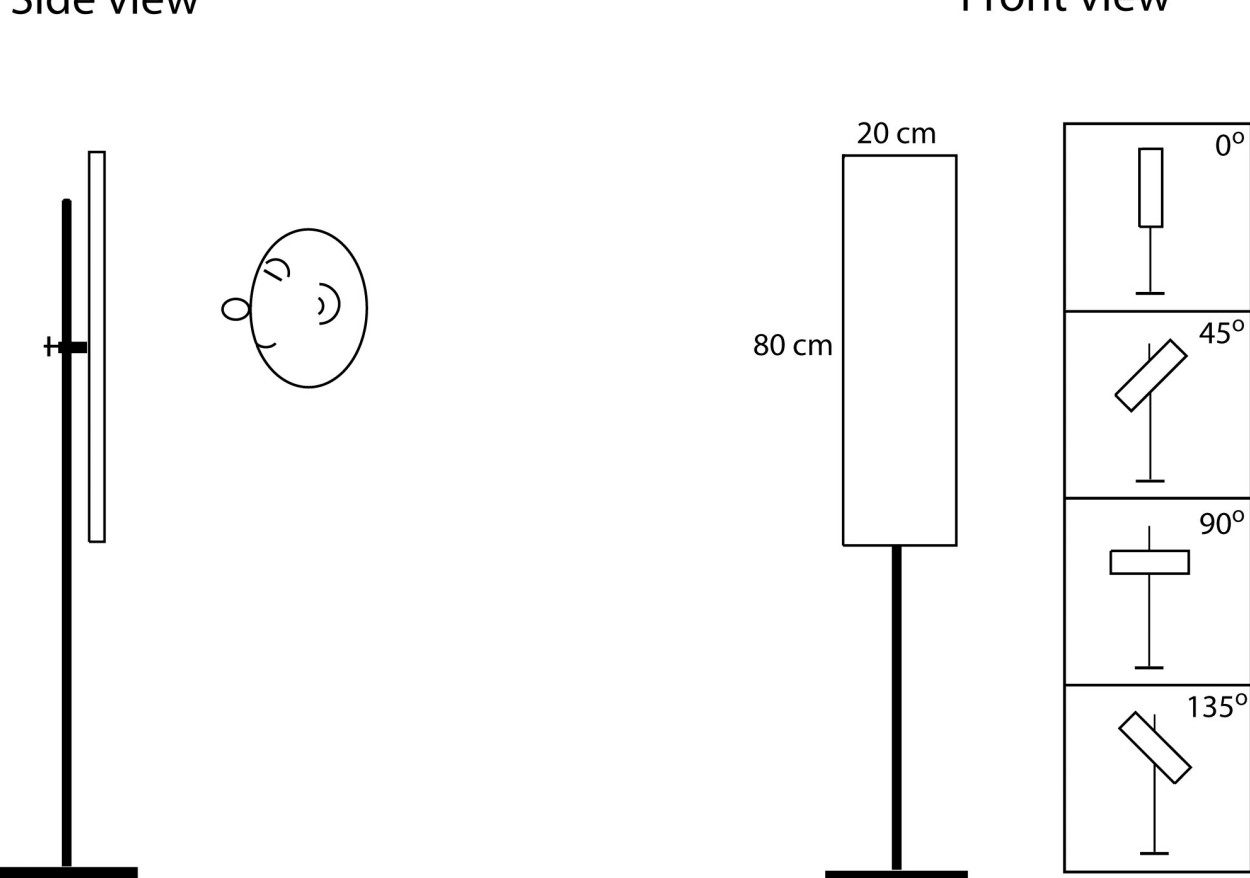

**Fig 3. Illustration of the apparatus used for the orientation perception task.** The inset on the left illustrates the different orientations the rectangle could be presented at, i.e. vertical, right side up (45˚), horizontal, or left side up (135˚), always facing the participant.

**Apparatus**: The same equipment as used for other computer-based tasks was used to play these sounds. Echolocation stimuli were presented at a level at which the highest intensity sound was approximately 80 dB SPL.

**Task and Procedure**: This task is a replication of the task introduced by [28] and details are described in that report. Briefly, binaural recordings of clicks and click-echoes were made with an anthropometric manikin in physical spaces comprising corridors in specific spatial arrangements (T-mazes, U-mazes, Z-mazes) that have been used previously by other labs to investigate navigation based on non-visual cues, e.g. [35]. Whilst visually the T-, U- and Z- mazes may appear similar to one another, and possibly easy to navigate, our results suggest that this is not the case for participants navigating these mazes using echo-acoustic cues in our paradigm—that is, they had to learn how to do this. Binaural recordings were then used to construct virtual spaces of corridors in those same spatial arrangements, through which participants could navigate using keys on a computer keyboard (see Fig 4). Participants listened to the click and click-echo recording (over headphones) corresponding to the location that they currently occupied and the orientation that they were facing. If participants pressed a key that would result in them walking into a wall, they would hear a 50-ms long 500-Hz tone telling them that this move was not possible (a collision error) and they would remain in the original location. On each trial, participants started at one of four starting positions facing into

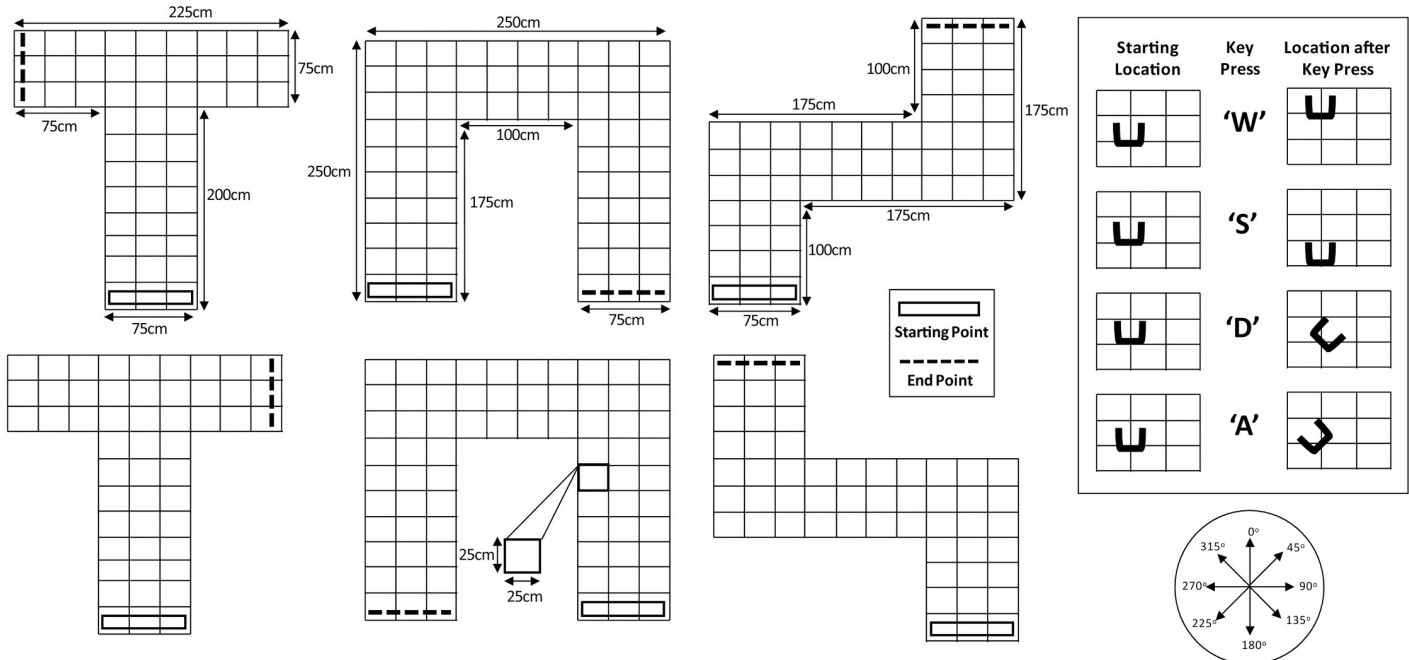

**Fig 4. Illustration of spatial arrangements used to construct virtual spaces (T-mazes, U-mazes, Z-mazes) for the virtual navigation task.** In all mazes, the box represents the starting area and the dashed black line symbolises the end point which sounded echo-acoustically different because it had been constructed from corrugated plastic sheets. For each position (i.e. intersection in each route) there were eight sound recordings (0˚–315˚ in 45˚ steps). To navigate, participants used the computer keyboard (inset on the right-hand side). Each press of the 'W' key would move the participant one step forward in the virtual maze and the 'S' key would move them one step backwards, but still facing in the same direction. Each press of the 'D' key would rotate the participant 45˚ in clockwise direction and the 'A' key would rotate them 45˚ anti-clockwise.

the maze, and they had 180 seconds to navigate to the end of the maze. The end of each maze was echo-acoustically distinct from the rest because it was constructed from corrugated plastic sheets, rather than poster boards. Each trial ended either when participants navigated to this goal (a "success") or when 180 seconds had passed (a "failure"). There were two versions of each maze which were mirror symmetric versions of one another (see Fig 4). In sessions 1–18, participants trained with one half of the spaces, and in sessions 19 and 20 were tested on both the trained and untrained (mirror symmetric) versions. This way we could determine to what degree the echo-acoustic skills they had learned would transfer to a new set of spaces in which they could not simply rely on rote behaviour to navigate the mazes. Participants were quasi-randomly assigned to which version of each maze they would start with. Sessions 1–18 each had 18 trials (6 repetitions of each of three mazes), and sessions 19 and 20 each had 36 trials (6 repetitions of each of the 6 mazes). Furthermore, from session 15 onwards we introduced an unpredictable starting orientation on each trial, as well as a 10-s timeout for each collision error. During the timeout no sounds would be played and the keys on the computer keyboard were unresponsive. EEs performed a single session matching the structure of sessions 1–14 but containing 36 trials (i.e. having 12 repetitions of each of three mazes).

*2.3.2.4. Echolocation outside the Lab in natural indoor and outdoor settings.* To facilitate that participants would use click-based echolocation in a novel environment outside the lab we also asked them to use echolocation to navigate outside the lab in a combination of indoor and outdoor settings for 20–30 minutes, either in the beginning or the end of a testing session, depending on participant preference that day. To this end, we asked participants to use mouth clicks to perceive the space around them, to orient themselves and to find their way. To avoid

apprehensiveness about wearing a blindfold whilst navigating in an unfamiliar public space, participants could choose to either wear a blindfold or close their eyes for this part of the training. All participants with normal vision and nine participants who were blind wore a blindfold. The remaining three (BC5, BC6, BC9) closed their eyes. During these sessions, every participant who was blind was also asked to use their long cane. Participants who were normally sighted were asked to also use a long cane, and they were instructed before the session in how to use it. All participants were encouraged to explore the environment in any way they wanted, including the long cane. It was explained to participants that the experimenter was available to guide them at any point should they wish to be guided, but participants were asked to still make mouth clicks and/or use the cane even when guided. The structure of the session was such that participants were invited to explore their environment starting from where they were when the session started (e.g. outside the lab or at the entrance to the psychology building) and where they wished to go. If the participant was undecided where to go, the experimenter made suggestions until the participant had an idea of what they wanted to do. The experimenter trailed participants at arm's reach to monitor their safety and asked them to make mouth clicks and to describe what was going through their mind (e.g. 'What made you stop walking?' or 'Why did you turn your head?' etc). The experimenter also answered participant questions (e.g. 'What is that in front of me?') or, if appropriate, the experimenter redirected the question and asked the participant to try and touch structures either with their hands or with the cane to find out what they were hearing.

The ambient noise present in these indoor and outdoor environments was not recorded, but all participants experienced a range of light to moderate to loud ambient sound levels in the environments that they sampled. Specifically, the environments ranged from empty to moderately busy to very busy corridors, classrooms, garden areas, communal areas, campuses, and outdoor pedestrian areas. Consistent with our observation that participants coped with these varied levels of ambient noise, we have also shown in laboratory based work [40] that intensity of click emissions can compensate for the presence of background noise in human echolocation.

### 2.4. Follow up survey

All BCs were contacted by phone or e-mail three months after having taken part in the research. They were asked a mix of yes/no and free text questions asking about the effects of having taken part in the research on aspects of their daily life and wellbeing. Questions asked are listed in Table 3. Apart from answering these specific questions, participants could also provide additional comments if they wished.

### 2.5. Data analysis

Our main statistical analyses across training tasks were conducted b model ANOVA with 'session' as a within subject factor and 'group' as a between subject factor, which were conducted

**Table 3. Details of questions asked in 3-month follow up survey.**

| Number | Question Text |
|---|---|
| 1 | Did taking part in the research improve your mobility? Y/N |
| 2 | If yes, in your own words how did it affect your mobility? |
| 3 | Did taking part in the research improve your independence? Y/N |
| 4 | If yes, in your own words how did it affect your independence? |
| 5 | Did taking part in the research improve your wellbeing? Y/N |
| 6 | If yes, in your own words how did it affect your wellbeing? |

with SPSS v26 (EEs were not included in these ANOVAs). Where sphericity was violated we report results that have been Greenhouse-Geisser corrected (GG). For independent samples t-tests, where the assumption of homogeneity of variances was violated we report results from a robust test (R) with adjusted degrees of freedom as implemented in SPSS. To compare data from EEs and SCs or BCs we used non-parametric Mann-Whitney-U tests since the sample of expert echolocators did not meet sample size requirements for parametric analyses. For the virtual navigation task, due to the differences in task structure introduced from session 15 onwards, data were analysed separately for sessions 1–14 and for 15–20. For analyses involving multiple comparisons, Bonferroni correction was applied.

## 2.6. Statistical power

We had practical limitations on our sample sizes, in particular for EEs and BCs. In order to demonstrate that we have sufficient sensitivity and specificity for our analyses, we calculated the minimum effect size that can be detected with our sample sizes. We did this separately for the three types of critical statistical tests that we use to support our main conclusions. These tests are: 1) testing if participants improved with practice, 2) testing whether there is a difference in performance between BCs and SCs, and 3) testing if learning is related to age. For all of these tests, we used G*Power 3.1.9.7 [41] to compute required effect sizes (for two-tailed tests), setting $\alpha$ to 0.05 and power to 0.8. Where required, correlation and non-sphericity correction values (GG) were estimated based on the data. Where G*Power computes effect sizes as Cohen's f, these values were converted to $\eta^2_p$ values as used in SPSS to be consistent with the units of our reported effect sizes, and for within-subjects effects calculations following [42]. Computed minimum effect sizes are reported throughout this manuscript alongside the observed effect sizes for any critical tests.

## 3. Results

### 3.1. Training tasks

**3.1.1. Control task (no click).** Minimum effect sizes for the main effect of 'session' and the interaction effect were .091, and for the main effect of 'group' it was .247. Performance (proportion correct) did not change significantly across sessions 1–20 ($F_{GG}$ (8.877, 213.04) = .743; p = .668; $\eta^2_p$: .030), as shown in Fig 5. There was no difference between participant groups (F (1,24) = 2.621; p = .119, $\eta^2_p$: .098) or interaction between group and session ($F_{GG}$ (8.877, 213.04) = .676; p = .728; $\eta^2_p$: .027). Participants' performance remained at chance level (.5) except in session 3, where it was significantly below chance (p<0.05 in a one-sample t-test).

The results of the control task suggest that any training effects we observed in other tasks (see subsequent results) were due to training in click-based echolocation and not due to improvements in passive echolocation or the use of ambient sound to perform the task, or other unspecific effects related to coming into the lab and taking part in the research.

**3.1.2. Size discrimination task.** We calculated two dependent variables as measures of performance in each session: the proportion of correct answers, and the distance at which participants performed the task (which increased following two consecutive sessions at proportion correct >.90). For proportion correct, minimum effect sizes for the main effect of 'session' and the interaction effect were .083, and for the main effect of 'group' it was .247. Proportion correct changed significantly across sessions ($F_{GG}$ (6.831, 163.955) = 9.312; p < .001; $\eta^2_p$: .280), and a significant linear trend is consistent with the idea that performance improved as training progressed (F(1,24) = 45.449; p < .001; $\eta^2_p$: .654). On average, proportion correct improved from .54 (session 1) to .79 (session 20). This pattern is also evident in the data shown in Fig 6 top panel. Based on the overall ANOVA analysis (not including EE group data), there

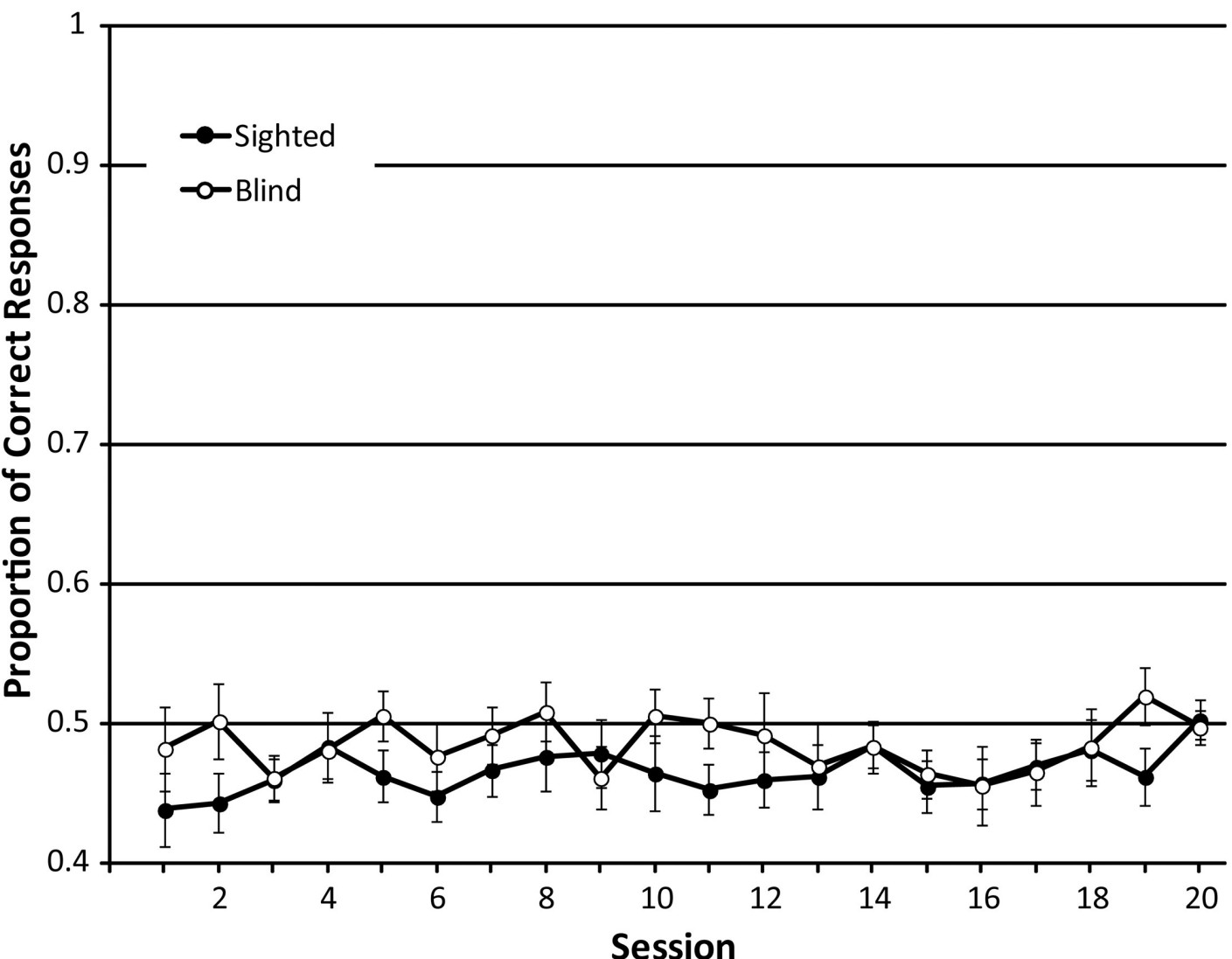

**Fig 5. Proportion of correct answers across sessions in the control training task.** Chance level for proportion correct is.5. Data from SCs and BCs are shown as black and white circles respectively, with each symbol representing the average and error bars representing the standard error of the mean across participants.

was no difference between participant groups (F(1,24) = 2.801; p = .107, $\eta^2_p$: .105) or interaction between group and session ($F_{GG}$ (6.831, 163.955) = 1.422; p = .201; $\eta^2_p$: .056). Participants' performance was not significantly different from chance (.5) in the first session, but was significantly better than chance in the second session and onwards (p<0.05 in a one-sample t-test). Focusing only on performance in the final session, SCs (mean accuracy: .84) performed better than BCs (mean accuracy: .74; t(24) = 2.417; p = .024). Furthermore, EEs (n = 3) performed significantly better (mean accuracy: .91) than SCs (mean accuracy: .84; U = 5; z = -2.042; p = .041) and BCs (mean accuracy: .74; U = 2; z = -2.311; p = .021) did in the final session, even though they performed the task at a different and greater distance (i.e. at 100 cm). In our paradigm, distance could only increase (and not decrease) depending on performance, i.e. the task could only get more difficult. Yet, our analysis of proportion correct data with ANOVA disregards changes in distance, which is equivalent to assuming that the task stays equally 'easy' at

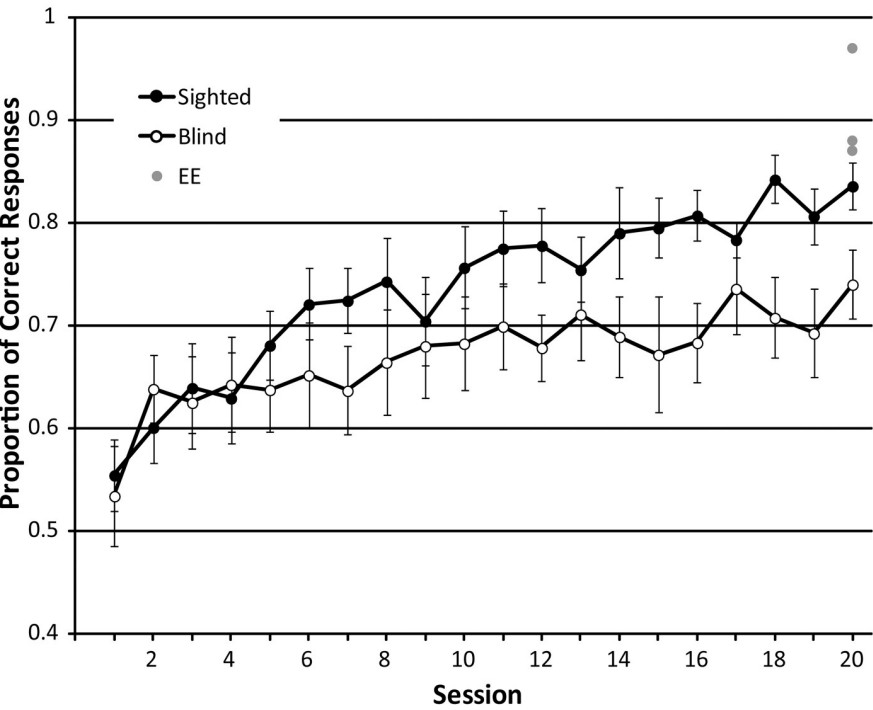

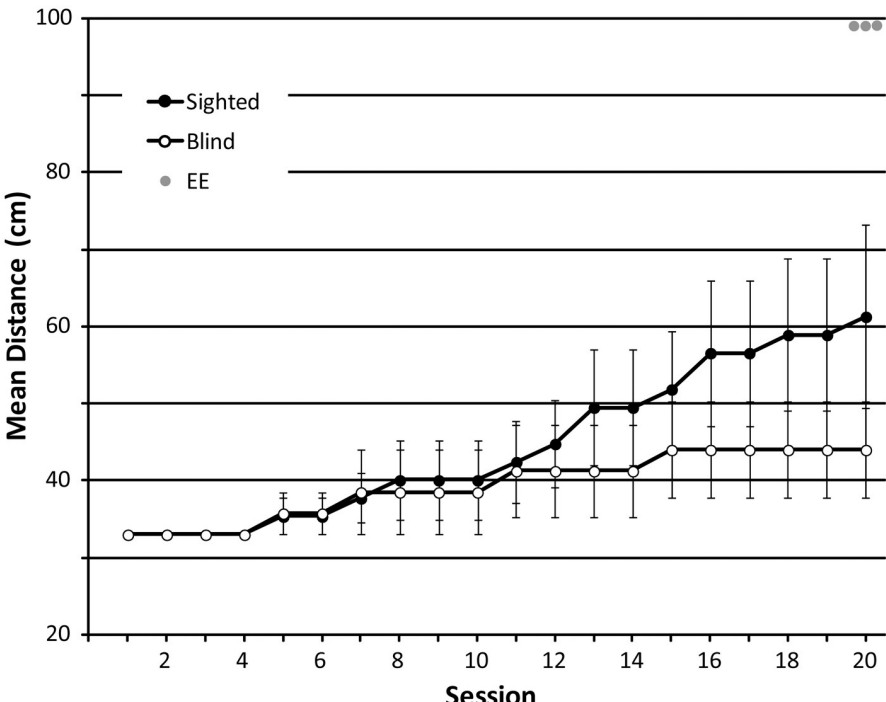

**Fig 6. Performance in the size training task.** Top panel: Proportion of correct answers across sessions. Chance level for proportion correct is.5. Bottom panel: Distance at which participants performed the task across sessions. Data from SCs and BCs are shown as black and white circles, respectively with each symbol representing the average and error bars representing the standard error of the mean across participants. Data from experts (n = 3) who completed only a single session without training and positioned at 100 cm distance are shown as grey circles. For comparison, they have been plotted at session 20.

all times. It follows that our ANOVA analysis gives a conservative estimate of participants' improvement, as it assumes that the task does not increase in difficulty with target distance.

For the distance that participants achieved in the task (our second measure of performance), minimum effect sizes for the main effect of 'session' and the interaction effect were .2, and for the main effect of 'group' it was .247. The ANOVA (not including EE group data) showed that the distance at which people performed the task changed significantly across sessions ($F_{GG}$(1.609, 38.618) = 7.270, p = .004; $\eta^2_p$: .232), and a significant linear trend is consistent with the idea that distance increased as training progressed F(1,24) = 9.331; p = .005; $\eta^2_p$: .28). This pattern is also evident in the data shown in Fig 6 bottom panel. This trend implies that participants' performance improved with training. Furthermore, there was no main effect of participant group (F(1,24) = .600; p = .446, $\eta^2_p$: .024) or interaction between group and session ($F_{GG}$ (1.609, 38.618) = 1.431; p = .250; $\eta^2_p$: .056). Focusing only on performance in session 20, there was no significant difference between SCs and BCs (t(24) = 1.224; p = .233). Additionally, because stimulus distance could only increase with training (and not decrease), we also conducted non-parametric analyses of these data (S1 File). These analyses further support our conclusions that SCs and BCs improved considerably with training, with no difference between groups.

**3.1.3. Orientation perception task.** These data were analysed in the same way as described for the size discrimination task. For proportion correct, minimum effect sizes for the main effect of 'session' and the interaction effect were .082, and for the main effect of 'group' it was .247. Proportion correct changed significantly across sessions ($F_{GG}$ (7.007, 168.169) = 9.882; p < .001; $\eta^2_p$: .292), and significant linear and quadratic trends are consistent with the idea that performance improved as training progressed (linear: F(1,24) = 39.581; p < .001; $\eta^2_p$: .623; quadratic: F(1,24) = 6.936; p = .015; $\eta^2_p$: .224). On average, proportion correct improved from .38 (session 1) to .69 (session 20). This pattern is also evident in the data shown in Fig 7 top panel. As with the size discrimination task, this is a conservative estimate of participants' improvement. Based on the overall ANOVA (not including EE group data), there was no difference between groups (F(1,24) = 2.967; p = .098, $\eta^2_p$: .110) or interaction between group and session ($F_{GG}$ (7.007, 168.169) = 1.086; p = .374; $\eta^2_p$: .043). Interestingly, participants' were able to do this task better than chance (.25) even without training (p<0.05 for all sessions, a one-sample t-test), but they still improved as sessions progressed. Focusing only on performance in the final session, there was no difference between SCs (mean accuracy: .76) and BCs (mean accuracy: .62; t(24) = 2.026; p = .054). Furthermore, although EEs (n = 5) did not perform significantly better than BCs did in the final session (U = 12; z = -1.901; p = .057), note that this effect is only marginally non-significant. EEs also did not perform better than SCs did in the final session (U = 21; z = -1.310; p = .190). Note also that EEs performed the task at a different and greater distance (i.e. at 100 cm) than the BCs and SCs did. Again, just as for the size discrimination task, our analysis of proportion correct data with ANOVA disregards changes in distance, which is equivalent to assuming that the task stays equally 'easy' at all times, and it follows that our ANOVA analysis gives a conservative estimate of participants' improvement.

For distance, minimum effect sizes for the main effect of 'session' and the interaction effect were .22, and for the main effect of 'group' it was .247. Based on the overall ANOVA (not including EE group data), the distance at which people performed the task changed significantly across sessions ($F_{GG}$(1.363, 32.704) = 8.746, p = .003; $\eta^2_p$: .267), and a significant linear trend is consistent with the idea that distance increased as training progressed F(1,24) = 10.274; p = .004; $\eta^2_p$: .30). This pattern is also evident in the data shown in Fig 7 bottom panel. Further to this, the ANOVA showed that there was no main effect of participant group (F (1,24) = .649; p = .428, $\eta^2_p$: .026) or interaction between group and session ($F_{GG}$ (1.363, 32.704) = .572; p = .506; $\eta^2_p$: .023). Focusing only on the distance achieved in the final session,

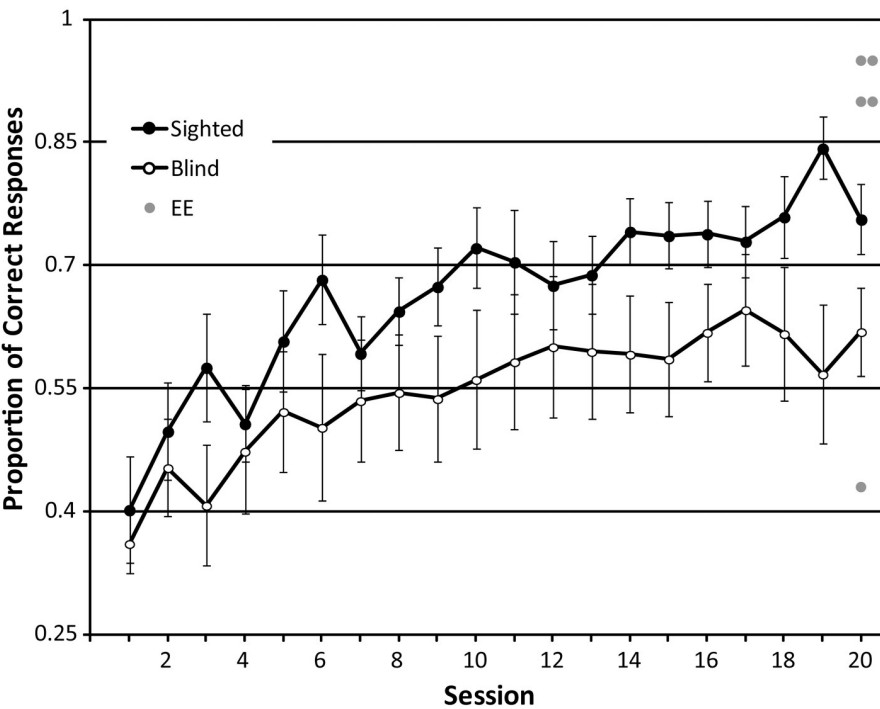

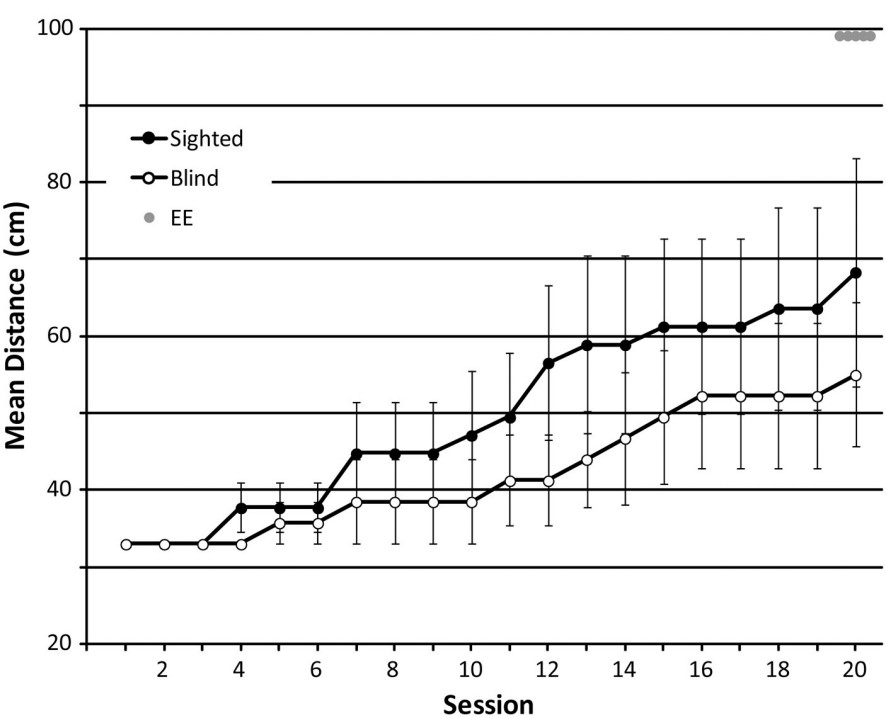

**Fig 7. Performance in the orientation training task.** Top panel: Proportion of correct answers across sessions. Chance level for proportion correct is.25. Bottom panel: Distance at which participants performed the task across sessions. Data from SCs and BCs are shown as black and white circles, respectively with each symbol representing the average and error bars representing the standard error of the mean across participants. Data from experts (n = 5) who completed only a single session without training and positioned at 100 cm distance are shown as grey circles. For comparison, they have been plotted at session 20.

there was no significant difference between SCs and BCs (t(24) = .731; p = .472). As with the size discrimination data, additional non-parametric analyses (S1 File) further support the conclusion that, as training progressed, participants were able to perform the task at farther distances, i.e., their performance improved, and this seemed to be the same for both BCs and SCs.

**3.1.4. Virtual navigation task.**  For each session in the virtual navigation task, we calculated each participant's mean completion time, mean number of collisions, and proportion of successful maze completions, in order to determine that any improvements in performance or group differences in performance apply to measures of both speed and accuracy. For sessions 1–14 we analyzed these data with mixed model ANOVA, with session (1–14) and group (BC, SC) as factors. The expected difficulty of sessions 15–20 was greater due to randomised starting locations and error timeouts, therefore we analyzed these sessions separately, with session (15–20) and group (blind, sighted) as factors. In sessions 19 and 20 people traversed both new (untrained) as well as old (trained) mazes. Any analysis investigating effects of session therefore only considered the mazes that people had trained with. We performed additional analyses to verify that the randomised starting locations and error timeouts did in fact lead to increased task difficulty (S1 File). Finally, to assess to what degree the skills that people had acquired generalized to novel virtual spaces, we compared performance between trained and untrained mazes with a mixed model ANOVA with novelty (old, new) as a within subject repeated factor and group (blind, sighted) as a between subject factor.

*3.1.4.1 Performance across repeated sessions*

**Time to complete maze**

For sessions 1 through 14, minimum effect size for the main effect of 'session' and the interaction effect was.102, and for the main effect of 'group' it was.247. Maze completion time changed significantly across sessions 1 through 14 ($F_{GG}$(3.474, 83.376) = 46.779; p < .001; $\eta^2_p$: .661), and significant linear and quadratic trends are consistent with the idea that participants became faster as training progressed (linear: F(1,24) = 85.635; p < .001; $\eta^2_p$: .781; quadratic: F(1,24) = 49.237; p < .001; $\eta^2_p$: .672). On average, maze completion time reduced from 119.30 s (session 1) to 48.42 s (session 14). This pattern is also evident in the data shown in Fig 8. The ANOVA showed no main effect of participant group (F(1,24) = 2.399; p = .135, $\eta^2_p$: .091) or interaction between group and session ($F_{GG}$ (3.474, 83.376) = 2.233; p = .081; $\eta^2_p$: .085).

Focusing only on completion time in session 14, there was no significant difference between SCs and BCs (t(24) = 1.502; p = .146). Furthermore, BCs did not significantly differ in completion time compared to EEs (n = 4; U = 17; z = -.849; p = .396), but SCs were significantly faster (mean time: 40.87 s) than EEs (mean time: 67.49; U = 9; z = -2.018; p = .044), although this effect was only marginally significant.

For sessions 15 through 20, minimum effect size for the main effect of 'session' and the interaction effect was.100, and for the main effect of 'group' it was.247. From session 15 through 20, completion time changed again significantly across sessions (F(5, 120) = 7.997, p < .001; $\eta^2_p$: .250), and a significant linear trend is consistent with participants becoming faster as training progressed (F(1,24) = 21.384; p < .001; $\eta^2_p$: .471). On average, completion time reduced from 110.21 s (session 15) to 90.63 s (session 20). This pattern is also evident in the data shown in Fig 8. Throughout these sessions, SCs were faster (mean time: 78.99 s) than BCs (mean time: 115.93 s; F(1,24) = 9.553; p = .005, $\eta^2_p$: .285), providing evidence that SCs were less affected by the error timeouts and random starting orientations compared to BCs (see also S1 File). There was no significant interaction between group and session (F(5, 120) = .615; p = .689; $\eta^2_p$: .025). To determine if the group difference could possibly be explained by the age (i.e. participants who were blind were older than participants who were sighted, see section 'Participants'), we included age as a covariate. With age as covariate, the effect of group became non-significant (F(1,23) = 3.687; p = .067; $\eta^2_p$: .138), although only marginally so, and

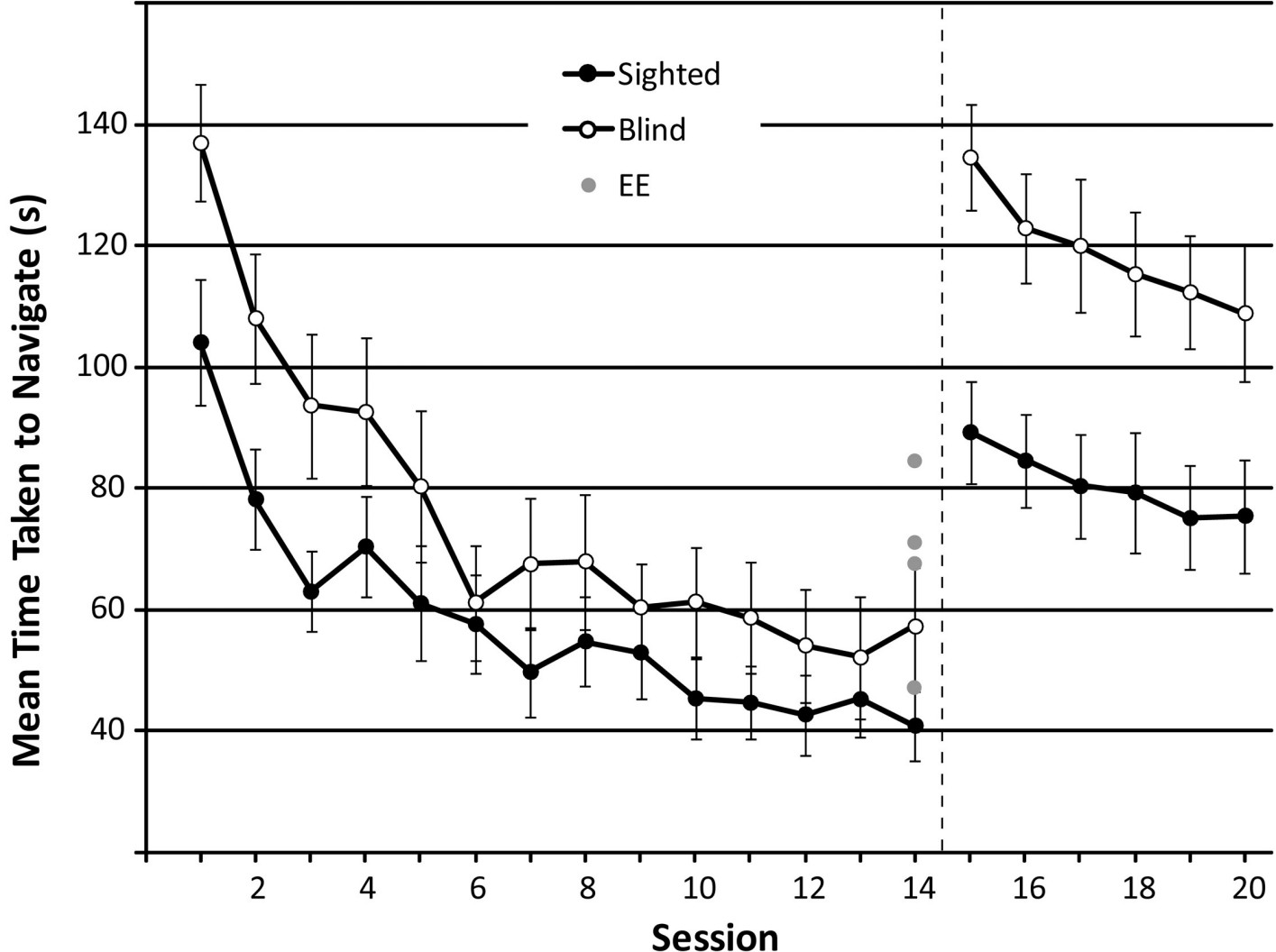

**Fig 8. The mean time taken (seconds) to complete various mazes in sessions 1–20.** In session 15, unpredictable starting orientations were introduced, along with a 15 s timeout when a collision occurred. This is represented by the dashed black line. Data from SCs and BCs are shown as black and white circles, respectively with each symbol representing the average and error bars representing the standard error of the mean across participants. Data from experts (n = 4) who completed only a single session without training are shown as grey circles. For comparison, they have been plotted at session 14.

the effect of age itself was significant (F(1,23) = 7.340; p = .013; $\eta^2_p$: .242), and positive (average standardized beta weight for the covariate: 2.51) suggesting that older age is associated with longer maze completion times, and that this might be responsible for differences in performance between SCs and BCs in this task.

**Number of collisions**

For sessions 1 through 14, minimum effect size for the main effect of 'session' and the interaction effect was.120, and for the main effect of 'group' it was.247. The number of collisions changed significantly across sessions 1 through 14 ($F_{GG}$(3.779, 90.697) = 6.753, p < .001; $\eta^2_p$: .220), and a significant linear trend is consistent with the idea that collisions became fewer as training progressed (F(1,24) = 29.399; p < .001; $\eta^2_p$: .551). On average, the number of collisions reduced from 6.57 (session 1) to 3.64 (session 14). This pattern is also evident in the data shown in Fig 9. Throughout these sessions, SCs had fewer collisions (mean: 3.85; SD: 3.18)

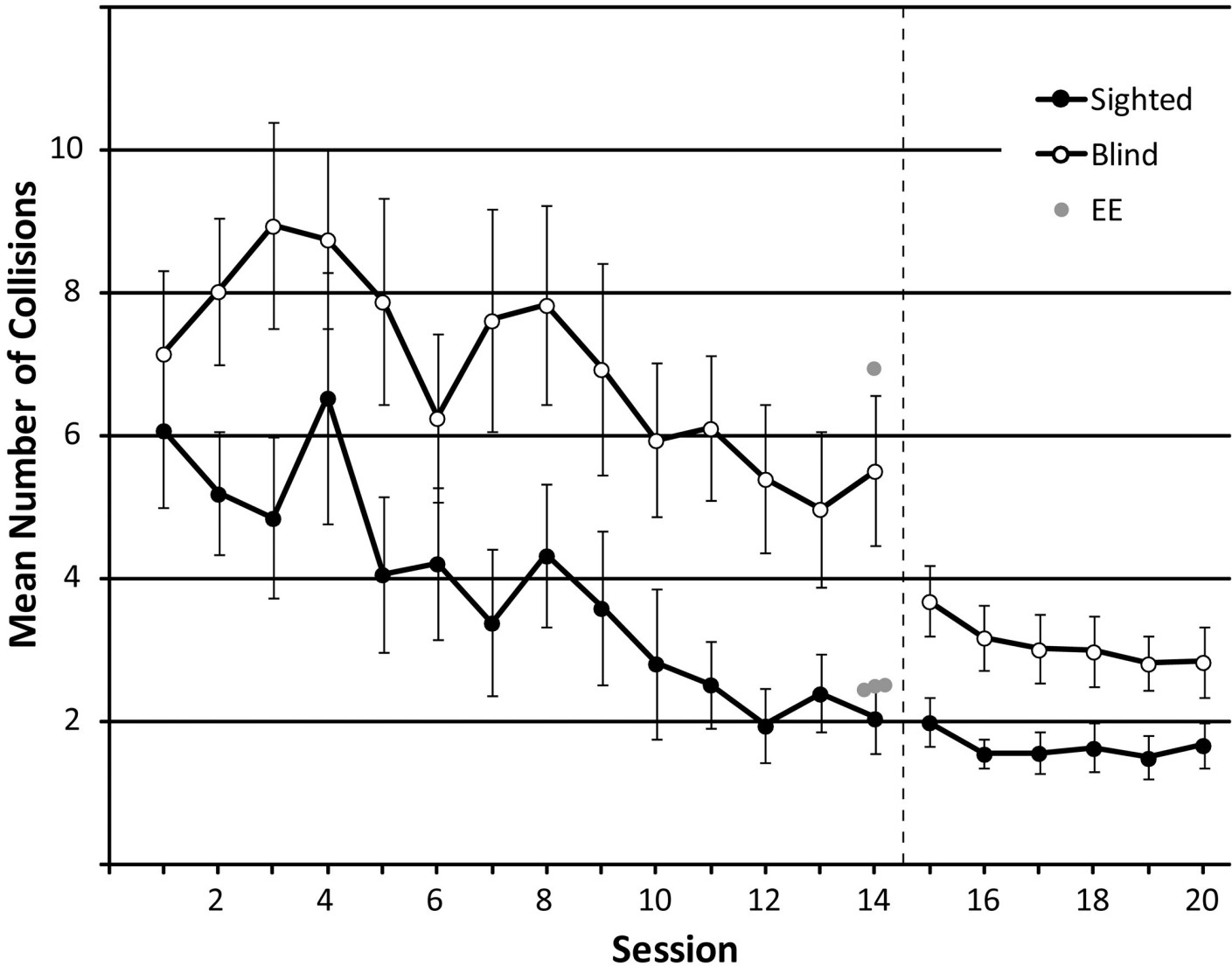

**Fig 9. The mean number of collisions made in sessions 1–20.** In session 15, unpredictable starting orientations were introduced, along with a 15 s timeout when a collision occurred. This is represented by the dashed black line. Data from SCs and BCs are shown as black and white circles, respectively with each symbol representing the average and error bars representing the standard error of the mean across participants. Data from experts (n = 4) who completed only a single session without training are shown as grey circles. For comparison, they have been plotted at session 14.

than BCs (mean: 6.95; SD: 3.18; F(1,24) = 6.138; p = .021, $\eta^2_p$: .204). The interaction between group and session was not significant ($F_{GG}$ (3.779, 90.697) = .761; p = .547; $\eta^2_p$: .031). With age included as covariate, the effect of group nonetheless remained significant ($F_{GG}$(3.722,85.6) = 3.004; p = .025; $\eta^2_p$: .116), and the effect of age itself was non-significant (F(1,23) = .421; p = .523; $\eta^2_p$: .018), suggesting that age differences do not underlie the difference in performance between BCs and SCs.

Focussing just on number of collisions in session 14, SCs performed significantly better (mean: 2.04) than BCs (mean: 5.51; t(15.836) = 3.006; p = .008), consistent with the significant main effect of 'group'. Furthermore, EEs (n = 4) did not differ significantly compared to either BCs (U = 11; z = -1.578; p = .115) or SCs (U = 12; z = -1.701; p = .089), suggesting that participants had learned to perform like experts by session 14.

For sessions 15 through 20, minimum effect size for the main effect of 'session' and the interaction effect was.100, and for the main effect of 'group' it was.247. The number of collisions changed again significantly across sessions (F(5, 120) = 5.659, p < .001; $\eta^2_p$: .191), and significant linear and quadratic trends are consistent with the idea that collisions became fewer as training progressed (linear: F(1,24) = 15.153; p = .001; $\eta^2_p$: .387; quadratic: F(1,24) = 15.290; p = .001; $\eta^2_p$: .389). On average, the number of collisions reduced from 2.77 (session 15) to 2.21 (session 20). This pattern is also evident in the data shown in Fig 9. Again, SCs had fewer collisions (mean: 1.650; SD: 1.298) than BCs (mean: 3.083; SD: 1.299; F(1,24) = 7.870; p = .010, $\eta^2_p$: .247) throughout these sessions, and there was no significant interaction between group and session (F(5, 120) = .950; p = .451; $\eta^2_p$ a: .038). With age included as a covariate, the effect of group became non-significant (F(1,23) = 3.445; p = .076; $\eta^2_p$: .13), but the effect of age itself was also non-significant (F(1,23) = 2.972; p = .098; $\eta^2_p$: .114), suggesting that whilst including age removes significant group differences, older age in itself is not associated with more collisions in this task.

**Proportion of maze successes**

For sessions 1 through 14, minimum effect size for the main effect of 'session' and the interaction effect was.118, and for the main effect of 'group' it was.247. The proportion of mazes successfully completed changed significantly across sessions ($F_{GG}$(3.889, 93.328) = 16.303, p < .001; $\eta^2_p$: .405), and significant linear and quadratic trends are consistent with the idea that people successfully completed a greater proportion of mazes as training progressed (linear: F (1,24) = 33.494; p < .001; eta: .583; quadratic: F(1,24) = 24.199;p < .001; eta: .502). On average, the proportion of successfully completed mazes increased from.62 (session 1) to.94 (session 14). This pattern is also evident in the data shown in Fig 10. There was no difference between participant groups (F(1,24) = 2.548; p = .124, $\eta^2_p$: .096) or interaction between group and session ($F_{GG}$ (3.889, 93.328) = 2.239; p = .096; $\eta^2_p$: .085). Focusing only on performance in session 14, there was no significant difference between BCs and SCs (t(12.839) = 2.207; p = .064), although this was only marginally non-significant. Focussing just on session 14, EEs (n = 4) did not perform significantly differently compared to either BCs (U = 22.5; z = -.187; p = .851) or SCs (U = 13.5; z = -1.836; p = .066), but again this was only marginally non-significant.

For sessions 15 through 20, minimum effect size for the main effect of 'session' and the interaction effect was.100, and for the main effect of 'group' it was.247. The proportion of mazes successfully completed changed again significantly across sessions (F(5, 120) = 4.382, p = .001; $\eta^2_p$: .154), and a significant linear trend is consistent with the idea that people successfully completed a greater proportion of mazes as training progressed (F(1,24) = 16.640; p < .001; e $\eta^2_p$ ta: .409). On average, the proportion of successfully completed mazes increased from.72 (session 15) to.81 (session 20). This pattern is also evident in the data shown in Fig 10. Again, SCs completed a higher proportion of mazes successfully (mean: .88) compared to BCs (mean: .66) throughout these sessions (F(1,24) = 7.612; p = .011, $\eta^2_p$: .241), with no significant interaction between group and session (F (5, 120) = 2.093; p = .071; $\eta^2_p$: .08). Including age as a covariate, the effect of group became non-significant (F(1,23) = 2.291; p = .144; $\eta^2_p$: .091), and the effect of age itself was significant (F(1,23) = 9.665; p = .005; $\eta^2_p$: .296), and negative (average standardized beta weight for the covariate: -3.222) suggesting that older age is associated with lower proportion of successful maze completions, and that this might be responsible for differences in performance between SCs and BCs in this task.

*3.1.4.2. Performance for 'Old' vs. 'New' Mazes.* A mixed model ANOVA with novelty (old, new) as within subject factor, and group (sighted, blind) as between subject factor was used to compare performance for 'old' (trained) and 'new' (untrained) mazes in sessions 19 and 20. We examined completion time, number of collisions made and proportion of mazes successfully completed. If the skills that people have acquired during training transfer to novel,

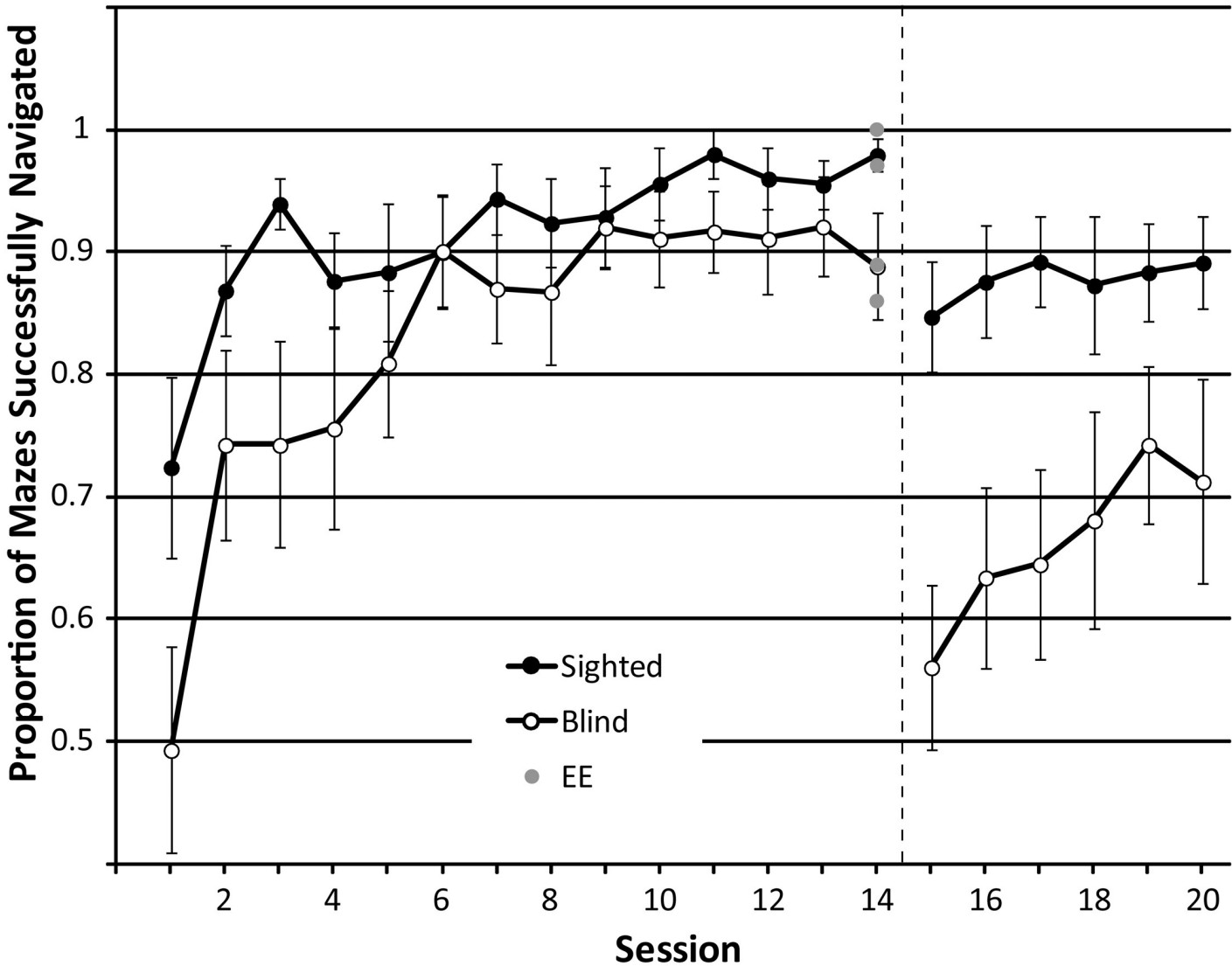

**Fig 10. The average proportion of successful maze completions in sessions 1–20.** In session 15, unpredictable starting orientations were introduced, along with a 15 s timeout when a collision occurred. This is represented by the dashed black line. Data from SCs and BCs are shown as black and white circles respectively, with each symbol representing the average and error bars representing the standard error of the mean across participants. Data from experts (n = 4) who completed only a single session without training are shown as grey circles. For comparison, they have been plotted at session 14.

untrained spaces, we should not observe any difference in performance between old and new mazes, i.e. no effect of novelty. Minimum effect size for 'novelty' and the interaction was.262, and for 'group' it was.247.

Completion time did not differ between 'old' and 'new' mazes (F(1,24) = 1.804; p = .192; $\eta^2_p$: .070). There was also no interaction between novelty and group (F(1,24) = .160; p = .693; $\eta^2_p$: .007), but as expected from the analysis across repeated sessions, SCs completed mazes faster (mean: 76.35; SD: 31.96) than BCs (mean: 112.56; SD: 31.96; F(1,24) = 8.298; p = .008; $\eta^2_p$: .257). This pattern is also illustrated in Fig 11A.

The number of collisions did not differ between 'old' and 'new' mazes (F(1,24) = 2.995; p = .096; $\eta^2_p$: .111). There was also no interaction between novelty and group (F(1,24) = .722; p = .404; $\eta^2_p$: .029), but as expected from the analysis across repeated sessions, SCs had fewer

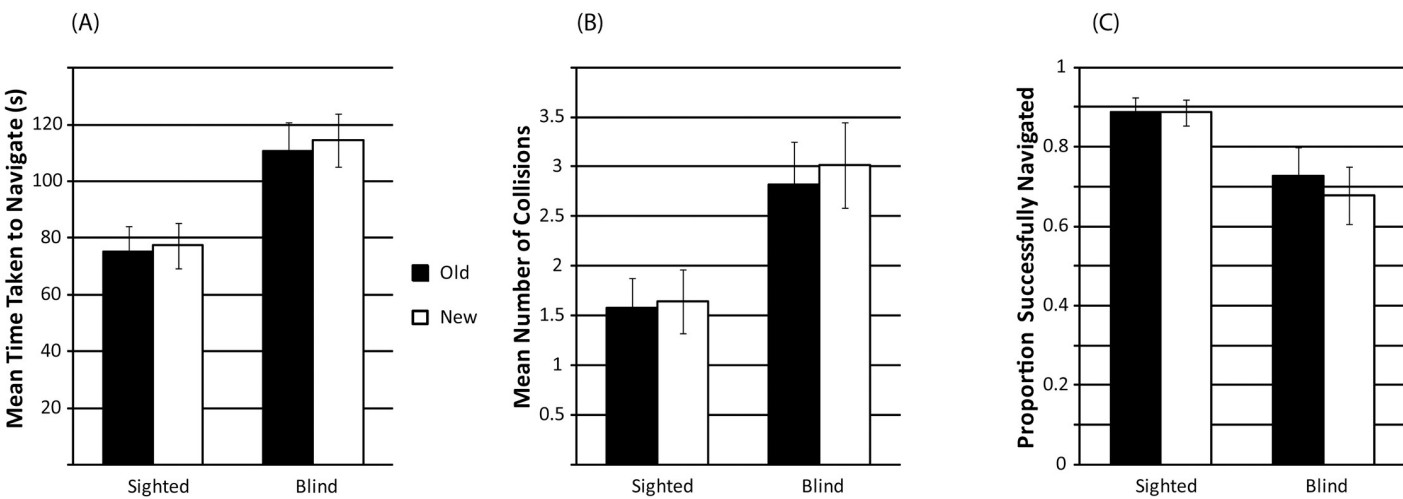

**Fig 11.** **(A)** Mean time taken (seconds) to navigate 'old' and 'new' mazes. **(B)**. Mean number of collisions made when navigating 'old' and 'new' mazes. **(C)**. Proportion of 'old' and 'new' mazes successfully navigated. Error bars represent the standard error of the mean across participants.

collisions (mean: 1.61; SD: 1.31) than BCs (mean: 2.92; SD: 1.31) ($F(1,24) = 6.412$; $p = .018$; $\eta^2_p$: .211). This pattern is also illustrated in Fig 11B.

The proportion of successes did not differ between 'old' and 'new' mazes ($F(1,24) = 2.592$; $p = .120$; $\eta^2_p$: .097). There was also no interaction between novelty and group ($F(1,24) = 2.215$; $p = .150$; $\eta^2_p$: .084), but as expected from the analysis across repeated sessions, SCs had fewer collisions (mean: .889; SD: .187) than BCs (mean: .70; SD: .187) ($F(1,24) = 6.250$; $p = .020$; $\eta^2_p$: .207). This pattern is also illustrated in Fig 11C.

## 3.5. The relationship between training improvement and factors of age, blindness and hearing sensitivity

To investigate how and if improvement during training is related to participants age', vision status or hearing, we calculated simple correlations and linear regression analyses. The minimum effect size was .51. First, for each of our training tasks and performance measures we calculated a single number to quantify participants' improvement by taking the difference in performance between their last and their first training session (for the virtual navigation task we used session 20 as last session). For each of these measures we then calculated simple correlations with age and blindness (dummy coded), and each of our six hearing measures (audiometric thresholds, binaural differences in audiometric thresholds, $DCI_{500}$, $DCI_{2000}$, $DFM_{500}$ and $DFM_{2000}$). We also ran calculations for SCs and BCs separately. We followed this up by calculating stepwise linear regression analyses where we determined if age and blindness (dummy coded), and our six hearing measures (audiometric thresholds, binaural differences in audiometric thresholds, $DCI_{500}$, $DCI_{2000}$, $DFM_{500}$ and $DFM_{2000}$) might account for a significant ($p < .05$) proportion of variance in performance.

For practical size and orientation tasks, participant improvement was not associated with age or blindness or with any of our hearing measures (max. r value = .373 for the whole group and max r value = -.482 for separate group analyses; all p values >.05). Fig 12 illustrates these results. The largest correlation ($r_{(N=12)} = -.482$) was observed for BCs in the orientation task and improvement in proportion correct, but this did not approach significance ($p = .112$).

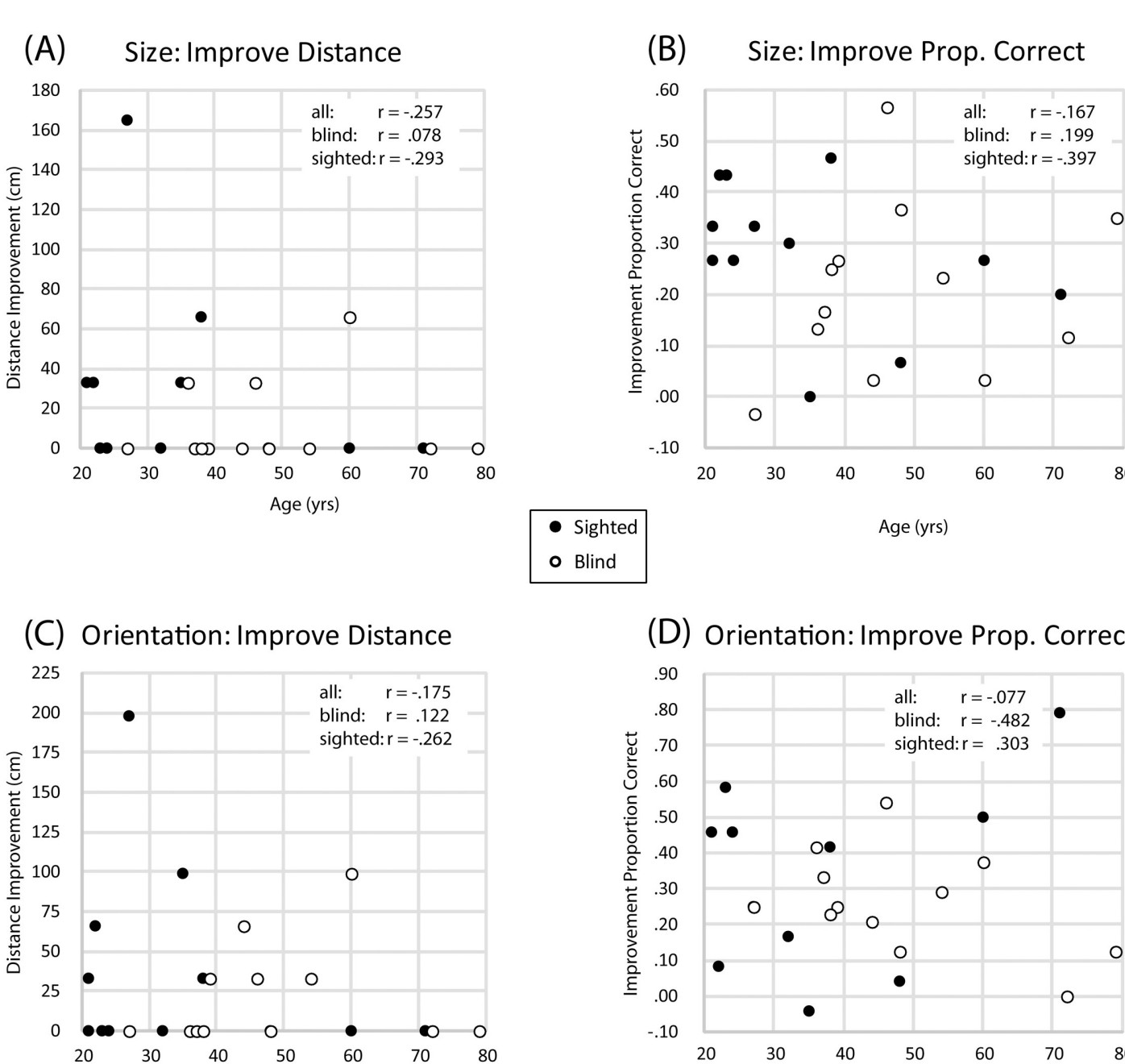

**Fig 12. Scatter plots of performance improvement in practical tasks plotted against age, split by participant group.** (**A**) Improvement in distance for the size task plotted against age. (**B**) Improvement in proportion correct for the size task plotted against age. (**C**) Improvement in distance for the orientation task plotted against age. (**D**) Improvement in proportion correct for the orientation task plotted against age. Data for BCs and SCs are in white and black symbols, respectively. Correlations are indicated in each plot. There is no relationship between age and/or blindness and performance improvement in practical tasks.

Overall, there is no evidence for any pattern relating age or blindness to performance in our practical tasks.

Considering our virtual navigation task, for improvement in proportion of successful maze completions (Fig 13A), there was a significant correlation with age only for SCs. For

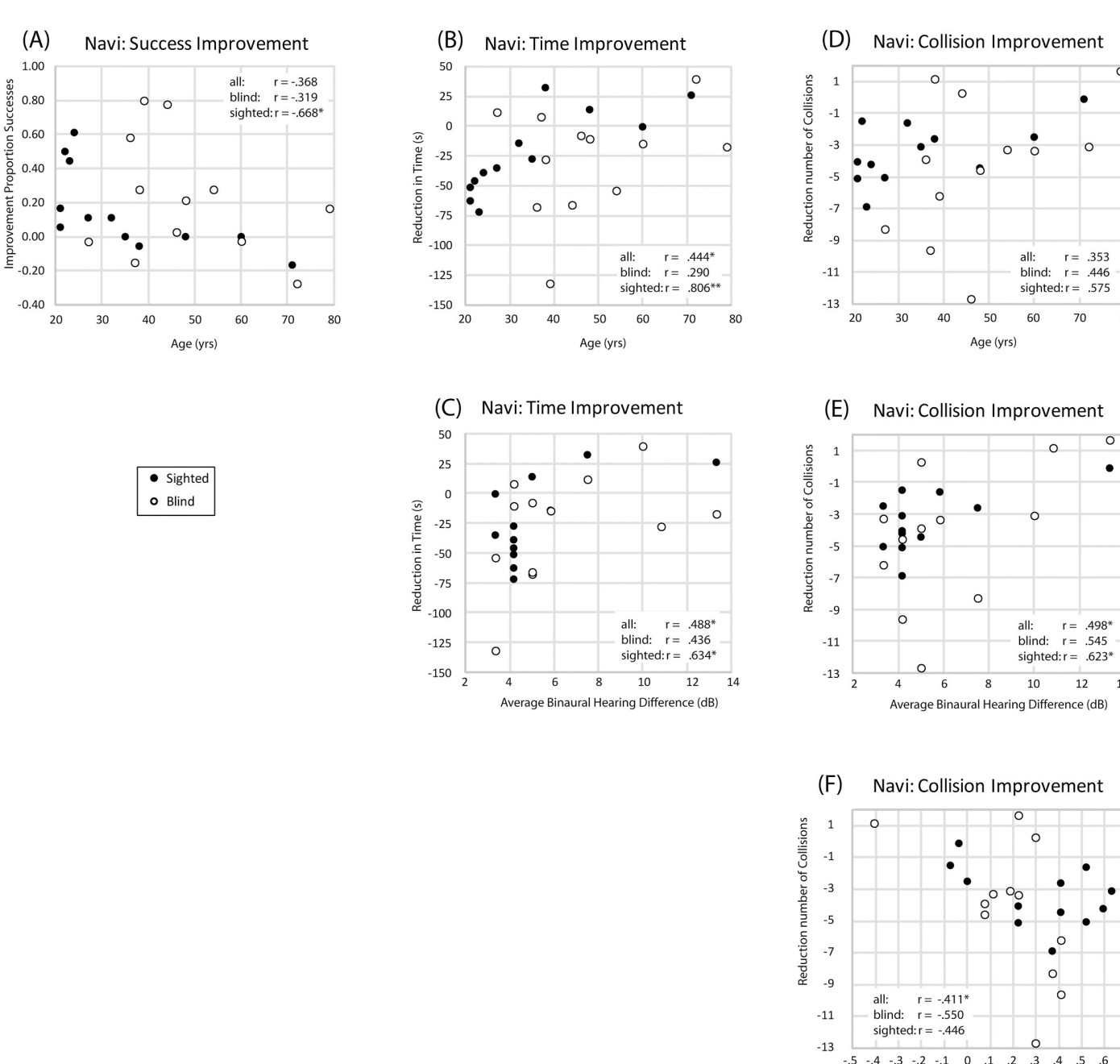

**Fig 13. Scatterplots for improvement in performance in the virtual navigation task.** Any significant correlations are indicated in each plot with an asterisk. **(A)** Improvement in successes for the virtual navigation task plotted against age. **(B)** Improvement in time taken to complete mazes for the virtual navigation task plotted against age. **(C)** Improvement in time taken to complete mazes for the virtual navigation task plotted against binaural hearing differences. **(D)** Improvement in number of collisions for the virtual navigation task plotted against age. **(E)** Improvement in number of collisions for the virtual navigation task plotted against binaural hearing differences. **(F)** Improvement in number of collisions for the virtual navigation task plotted against performance in the DFM test at 500 Hz. Data for blind and sighted participants are in white and black symbols, respectively. Correlations are indicated in each plot, with significance indicated as * = p < .05; ** = p < .01.

improvement in completion time (Fig 13B), there was a significant correlation between performance and age for SCs, and also when SCs and BCs were considered together (Fig 13B). For improvement in the number of collisions, there were no correlations with age (Fig 13D).

The remaining panels (Fig 13C, 13E and 13F) illustrate remaining significant correlations between improvement in performance and hearing measures. Specifically, participants' improvement in completion time was correlated with their average absolute binaural differences in hearing thresholds for both SCs, and when SCs and BCs were considered together (Fig 13C). Furthermore, improvement in the number of collisions was significantly correlated with participants' average absolute binaural differences in hearing thresholds for SCs and for the whole group (Fig 13E), and with their performance in the DFM test at 500Hz for the whole group (Fig 13F).

To determine statistically which of these relationships explains most of the variance in performance, we ran stepwise linear regression analyses. These showed that both participants' improvement in completion time and number of collisions were significantly related to participants' average absolute binaural differences in hearing thresholds, but none of the other variables (including blindness) contributed significantly above and beyond binaural hearing differences. Specifically, 24% of the variance in the improvement in completion time between session 1 and 20 was predicted by binaural hearing differences ($F(1,22) = 6.895$; $p = .015$) and the relationship was positive so that smaller binaural differences were associated with larger reductions in completion times (standardized beta: .488, $t(22) = 2.626$; $p = .015$). Furthermore, 25% of the variance in the improvement in number of collisions between session 1 and 20 was predicted by binaural hearing differences ($F(1,22) = 7.273$; $p = .013$) and the relationship was positive so that smaller binaural differences were associated with larger reductions in collisions (standardized beta: .498, $t(22) = 2.697$; $p = .013$).

In sum, for any of the practical tasks there is no evidence for relationships between improvement in performance and blindness or age. For some aspects of the virtual navigation task (i.e. time taken to complete and number of collisions made), there is evidence that improvement in the time taken to complete the mazes is significantly related to age, but overall performance improvements are better explained by differences in binaural hearing. Binaural differences, however, were correlated with age, so that older age was associated with larger binaural differences ($r = .575$; $p = .003$), making their effects confounded and difficult to assess separately.

## 3.4. Effects of the training on mobility, independence and well-being outside the lab (3 months follow-up)

An important consideration is whether learning these echolocation skills is associated with real-life improvements in people's mobility, independence, and/or well-being. Thus, we here took the approach to follow up with participants who are blind 3 months after having taken part in the research and to ask them if and how the training in click-based echolocation affected their mobility, independence and wellbeing in their daily lives. All 12 blind participants took part in our follow up survey. In some cases participants responded with 'it' to refer to click-based echolocation and its uses. Because quotes were transcribed verbatim from conversations on the phone (except for one participant who responded via email), clarification has been added by the experimenter in parenthesis. All responses are provided in S2 Data.

12 out of 12 participants reported improved mobility, and the predominant theme was that the training in and current use of click-based had given them improved spatial perception which helps with navigation and moving about. Some free text comments are below.

**BC11**: *I feel that it (i.e. click-based echolocation) has improved my mobility in that I can pinpoint better a particular area that I want to be at and it (i.e. click-based echolocation) helps working me out how to get to that area, for example the route to a door*

**BC7**: *Helps with spatial awareness; when I am out it (i.e. click-based echolocation) is useful for example for finding landmarks and side streets.*

**BC12**: *I use the clicks when I am in the dark or in strange places; it (i.e. click-based echolocation) helps me detect obstacles or doors and to orient myself; I don't knock into things.*

**BC4**: *Helps me get a better impression of what is around me; It (i.e. click-based echolocation) helps me get my bearings.*

**BC5**: *I am better using audio feedback (i.e. click-based echolocation) whilst navigating busy environments such as shopping malls.*

**BC2**: *I use the clicks and echoes to find doors and entrances, to determine what is around; I use it (i.e. click-based echolocation) more in quiet environments, it is less useful for me in noisy places.*

**BC3**: *Improved spatial perception. (using click-based echolocation) I can tell if something is coming up in front of me, I can tell gaps, and I have a much better sense of where I am.*

*For example when I get out of a taxi, and the dog is not ready/not on harness yet, I can now use the clicks to find the gap in between parked cars, get on the footpath and get ready. This used to be something that I struggled with in the past, and that would worry me when taking a taxi because the drivers always say that yes we are right at the footpath, but often there is a row of cars in between the taxi and the path.*

*Or at home, we have a kitchen cupboard that when it is left open blocks the way from the living room into the kitchen, and I used to bang my head against that door when it was left open. Now I use the click and I can just hear it. Since using the clicks I have not banged my head on that cupboard door.*

*Also, when I walk around where I live I now have a much better sense where I am, for example where I live there is a small area with shops and there are steps in front of them and when I hear the steps (i.e. using click-based echolocation) I know where I am already from a distance.*

**BC10**: *Improved my mobility inside and outside. For example, inside I now walk without trailing walls or furniture. I can walk with things in my hand and I use echoes (i.e. click-based echolocation) to locate doorways and also if for example cupboard doors are open or shut. When I hoover, I use it (i.e. click-based echolocation) to find the hoover again; taking rubbish out around the corner and locating the bins etc.*

*It (i.e. click-based echolocation) has improved my perception and mobility so much that I also got to stage with my guide dog that I never thought I would get to, but I did!*

*For example, as I am walking along I can hear roadways, finding entrances, find bus shelter, to take travel with the train, parking meters; people are still hard to detect;*

*I am more confident, more independent; I am going to things that I did not do before; before I used to get really anxious to go anywhere by myself, but now I just go. I am less worried;*

*Also, my balance has improved, and I am quicker on my feet.*

**BC8**: *Improved directly in terms of indoors and outdoors in terms of echo use (i.e. click-based echolocation), but also my general use of sound has improved e.g. traffic. I have better safety and navigation skills.*

*I am now able to walk in a straight line—before I was veering a lot which is dangerous because it got me onto the road and into traffic at times.*

10 out of 12 participants (83%) reported improved independence, and the predominant theme was that the improved mobility enables them to do things by themselves. Those two participants who had answered negatively to the question had commented that they had been independent even before taking part, which is why they did not report an improvement. Some free text comments are below.

**BC11**: *I am capable of going to new areas and find out what is around me; I can get more detail more information without the cane, i.e. from a distance*

**BC7**: *I was independent before in terms of doing things by myself etc. but I think the better I get at it (i.e. click-based echolocation) which I still do as I am using it (i.e. click-based echolocation) every day, the more it will improve independence, e.g. for travelling to unfamiliar places alone.*

**BC12**: *Because of better mobility I am more independent.*

**BC9**: *I now go to places by myself; I prepare and walk by myself; Before the research (i.e. using click-based echolocation) I always needed someone else to come with me I could never go alone, but now I can do this by myself.*

**BC1**: *I am now far more confident that I can stay in my own home, even if I do get old alone. And that means a tremendous amount to me.*

*It (i.e. click-based echolocation) allows me to do things on my own. I do not need to rely on my partner; I can go by myself and explore new places. My partner understands that he needs to trust me to get myself about alone, and has promised to try and do that more in the future. Just for examples: Today I went to xxx on the bus by myself. I can go alone to my allotment now. I now do the ironing by myself—before my partner was too worried about me to let me do it by myself.*

**BC10**: *I feel that I am able to do more; if I want help I can ask but if not I can do it myself; for example we have an open plan office; there are good echoic landmarks; before other people used to guide me, now I just go and do it myself (i.e. by using click-based echolocation);*

*It (i.e. click-based echolocation) has also changed how I interact with people; I am more confident; people approach me more, including children.*

**BC8**: *(using click-based echolocation) In conjunction with long cane going shopping and to library by myself a; going to GP by myself; before I always needed help by another person.*

**BC5**: *The reason why I felt unable to answer yes to the above questions is due to the fact that having worked with several mobility and rehabilitation instructors over the years, I felt confident that all the methods and techniques I have learned over time work well enough to help me feel confident to get out and about safely and independently, even before I took part in the research.*

10 out of 12 participants (83%) reported improved wellbeing, and common themes were that through the training people realized that they can do this and that they are capable. Some participants also reported positive effects on relationships to other people including but not limited to family. Those two participants who had answered negatively to the question commented that they had been well before taking part, which is why they did not report an improvement. Some free text comments are below.

**BC11**: *It has given me a new vision for the future, new horizons of what is possible. It has improved my knowledge about my own capabilities as a human being*

**BC7**: *Given me an extra facet to life. I feel it is a bit like learning a new language. It was like setting a challenge to improve and keep improving at it (i.e. click-based echolocation).*

**BC6**: *More confident. I feel that I have another strength now (i.e. click-based echolocation).*

**BC12**: *I feel safer, more relaxed.*

**BC1**: *I have more confidence in myself. The training was challenging both mental and physical and I enjoyed that challenge. It was encouraging for me. I feel more alive.*

**BC3**: *I used to be worried that when my guide dog retired I would have a period of time where I could be unable to get about; this is what it was in the past when my dog retired; it made me very worried; now I can do it myself; I am actually considering not re-applying for a guide dog right away because I am doing just fine by myself.*

*I did not think that I would be picking this (i.e. click-based echolocation) up so well. I feel that I have really achieved a lot, and this makes me feel good.*

**BC10**: *I feel more awake and more alive; I feel more in control; if I go into an unfamiliar environment I persevere more; it (i.e. click-based echolocation) makes me feel more confident; at work meetings I find that now I can stay on much longer and I do not drift off.*

*I can do more with my children which makes me feel a better parent.*

*I don't get stressed as easily as before; I am more analytical, if needed I just solve the problem. My mother is confident to let me go; she has more confidence in me and says that she just leaves me to get on with it; it takes the pressure of her and makes for a much more mature relationship.*

**BC8**: *It (i.e. click-based echolocation) has not only given me better mobility skills and thus more independence but also the confidence to be able to do it, confidence in my abilities. For example, I have joined a new group that does things related to music. Before the research I would not have done it, because I would not have been confident enough that I can go to new places by myself.*

*Now I know that I can do it*!

**BC2**: *I was positive about my life etc. before the training.*

In sum, having taken part in our 10-week training program has made a positive impact on the lives of participants who are blind in terms of mobility, independence and well-being.

## 4. Discussion

Ovesrall, both sighted and blind participants showed clear improvements in echolocation ability across a range of practical and virtual tasks in a 10-week training program. In fact, every blind control participant improved with training. Most importantly, our blind participants also reported in a follow-up survey that learning these skills made a positive impact on their mobility, independence and well-being.

To summarise performance on the behavioural tasks, in the task of size discrimination, both BCs and SCs improved considerably with training, although they did not quite reach the performance of EEs. In the task of orientation perception, BCs and SCs again improved considerably with training and, in fact, their performance at the end of training matched that of EEs. In the virtual maze navigation task, BCs and SCs improved considerably on all three measures—completion time, number of collisions, and proportion of mazes successful navigated —implying that participants did not sacrifice accuracy for speed (or *vice versa*) as training progressed. Furthermore, in terms of maze completion time and number of collisions, BCs and SCs performed comparatively to EEs, and in one instance even showed superior performance (SCs had lower completion times than EEs). The overall good performance of EEs on all our tasks (without training) suggests that these tasks are sensitive to click-based echolocation abilities related to echolocation expertise, and that they therefore have good ecological validity. Furthermore, the fact that SCs and BCs were able to perform comparatively to EEs on some measures at the end of training suggest that the structure and length of our training schedule were sufficient to bring about remarkable changes in participants' echolocation abilities.

For our practical echo training tasks, there was a noticeable trend that SCs performed overall better than BCs throughout these tasks, and we did find that SCs performed significantly better than BCs in the final session of the size discrimination task. For our virtual navigation task, we also found that SCs performed significantly better on some measures, and this was more evident for the later sessions in which error timeouts and random starts were introduced. Our secondary analyses on the nature of these group differences shed some light on the possible underlying causes. Specifically, analysis of covariance suggested that these group differences could possibly be explained based on participants' age—our SCs were, on average, younger than our BCs. This is consistent with evidence that younger adults learn a computer-

based navigation task using a sensory substitution device better than older adults [35], possibly due to experience with computer game-like tasks (e.g. for review see [43]).

Importantly, when we quantified to degree to which participants improved from session 1 to session 20 in their abilities across each of the tasks, there was no evidence for an association between age and performance in the practical tasks, suggesting that age is unlikely to be a limiting factor in learning these echolocation abilities. We did find, however, that younger age was associated with greater improvements in the time it took people to navigate virtual mazes. Thus, even though our sample was small, the tasks and measures we used showed differences in their sensitivity to age, with active tasks showing no effects and computer-based tasks showing age effects with respect to completion times. Given our small sample, however, further research with larger samples and thus greater statistical power should continue investigating the role played by age for learning a novel sensory skill like echolocation. It is not possible to identify a single specific perceptual or cognitive factor that might underlie this association, but we did find that binaural differences in hearing, which were correlated with age in our sample, nonetheless explained more variance in learning than age did. Specifically, participants with smaller binaural hearing differences improved most in training. This is consistent with a previous report showing that people with better binaural hearing performed better in a computer-based echo-detection task [37], suggesting that differences in binaural hearing might be a limiting factor for participants learning computer-based echolocation skills. Future research should aim to replicate this result in a larger sample. Importantly, the effect we found did not play a role for active echolocation and real-life implications assessed through a survey, suggesting that any future research needs to assess these separate aspects carefully. There are other possible reasons that might explain performance differences between our BCs and SCs beyond blindness *per se* that we did not address in this study. For example, even though all our BCs were independent travellers, it is still possible that our SCs enjoyed greater levels of independent mobility compared to our BCs. In fact, our survey results to a degree attest to this in particular before taking part in the research, since the majority BCs point out that before taking part they relied more on other people in unfamiliar or novel situations. This in turn might manifest as a group difference in how people approach a novel training situation, which in turn may affect learning. We did not take measures relating to such factors at the beginning of our training, thus we cannot establish or rule out their effects, but it is nonetheless possible that SCs might have an advantage in our study for reasons beyond blindness and/or age. Although it is difficult to identify the specific reasons for any group differences, what is clear in our results is that both sighted and blind people can improve considerably in their echolocation ability with training.

It is important to assess the degree to which any improvements in ability are specific to click-based echolocation. Firstly, we did not find any effects of echolocation training on performance in our control task that did not require participants to make any clicks. This suggests that any effects we observed were specific to training in click-based echolocation, and not due to unspecific training effects on passive echolocation abilities or other strategies that could be the result of people visiting the lab on a regular basis and taking part in research. Secondly, in sessions 19–20 of the virtual navigation task we measured the degree to which participants were able to navigate unfamiliar mazes (i.e. those not experienced during training). We found that participants did this very well, with no difference in performance between the unfamiliar and familiar mazes, therefore suggesting that improvements were not limited to rote-based learning, and did indeed relate to echo-acoustic skills that could be used equally well in both the unfamiliar and familiar mazes. Furthermore, there was no difference between BCs and SCs in this generalisation from trained to untrained mazes, lending further support to our

conclusion that blindness/sightedness is not a strong predictor of the degree to which people learn echolocation skills.

Our findings that long term visual sensory deprivation in our BCs did not put them at an advantage for learning click-based echolocation (if anything, our SCs performed better) was surprising. This is because of the large literature suggesting increased neuroplasticity in the context of visual sensory deprivation [1–8], based on which we might have expected BCs to perform better. All participants who were blind had vision loss present from birth, even though two of them received their official diagnosis at an age that might be close to or beyond the onset of puberty. Thus, the majority of our participants would be classified as early blind. There is evidence to suggest that early and late onset blindness affect neuroplastic changes [8], so that future research should investigate how the results we found here might be affected by age at onset of blindness. Based on our findings that sighted participants learned well, we would expect that people with late onset of blindness would learn well also.

We had also asked participants about their echolocation use outside the lab, and all our participants who are blind did report in a follow-up survey that the training improved their mobility. 83% also reported that it positively affected their independence and wellbeing. All of our participants who were blind were competent long cane users and/or guide dog users prior to taking part in our training. Thus, everyone had previously already received training in orientation and mobility, and the effects that they experienced were on top of their already existing skills. With respect to mobility, the improvements that people experienced were all specific to the use of echolocation, i.e. the use of clicks and click echoes. These improvements then also affected independence and wellbeing, but people also reported that the effect of having acquired a new skill in itself provided a benefit to their wellbeing: They felt more capable. We did not run a separate study with a control task, i.e. training in another skill, such as for example a sensory substitution device that could also aid spatial sensing. As such, it is unclear if training with such a device would lead to similar benefits. But it is very clear from the data that training in echolocation provides real-life benefits to people who are blind. This extends the results of a previous study [31], which had used a correlational analysis to show that these two factors were related.

In conclusion, our results show that, somewhat surprisingly, blindness and age played only minor roles in the learning of echolocation, in particular for practical tasks, and training led to remarkable behavioural changes for all participants. Whilst there was a trend for sighted participants to perform better, most noticeably in our virtual navigation task, this is possibly explained by age, and/or binaural hearing measures related to age. Importantly, virtual navigation training effects still generalized equally for both groups, suggesting that even if blindness, age and/or binaural hearing differences may affect performance and learning, these do not impair generalization of what has been learned. Further research is needed to find out about underlying mechanisms. It is also important to quantify the degree to which abilities acquired in computer-based tasks translate into real-world settings. The results from our virtual navigation task establish that performance generalises very well from untrained to novel environments regardless of the person's sightedness, but we have not quantified how well these skills transfer to realistic scenarios in the physical world. Whilst our survey results demonstrate that the training as a whole (i.e. including both computer-based and practical tasks) affects realistic scenarios in the everyday life of blind individuals, we did not investigate if and how training of only the computer-based virtual navigation task would transfer into real life. This should be an important avenue for subsequent research.

Our study and results have implications for health and rehabilitation professionals working with people with vision impairments, in particular for people working in the field of orientation and mobility. Specifically, our survey demonstrates that all our participants who are blind

experienced positive effects on their everyday life. Thus, also on a practical level our data are important because they demonstrate that training click-based echolocation is useful for people with vision loss, and that people can learn to echolocate regardless of age or visual status. Based on our data we therefore suggest that any time is a good time to start learning click-based echolocation, and that it would make good sense to provide training in this skill as part of orientation and mobility instruction to people with vision impairments. Echolocation is currently not taught as part of mobility training and rehabilitation for blind people, and there is the possibility that some people are reluctant to use click-based echolocation due to a perceived stigma around making the clicks in social environments. Yet, blind people who use echolocation do so in a way that is adaptive to the social situation [17] and, as shown by our survey results, people new to echolocation are confident to use it in situations with other people. The potential barriers relating to perceived stigma are, therefore, perhaps much smaller than previously thought. Furthermore, it would also make sense to provide information and training in click-based echolocation to people who may still have good functional vision (e.g. like the sighted people in our study), but who are expected to lose vision later in life because of progressive eye conditions.

## Supporting information

**S1 Data. Excel table containing data from all participants in all tasks and measurements.** Each row contains data for one participant. Column headings indicate which data are shown. (XLSX)

**S2 Data. Excel table containing data from all blind participants in the survey.** Each row contains data for one participant. Column headings indicate which data are shown. (XLSX)

**S1 File. Word Document containing additional descriptions of DFM and DCI hearing tests, additional results from non-parametric analyses applied to distance data from size and orientation tasks, and additional results from analyses of virtual navigation task performance in session 14 and 15.** (DOCX)

## Acknowledgments

We thank all our participants for taking part. We thank The Durham talking Newspaper and the Sunderland and County Durham Royal Society for the Blind, and Guide Dogs for the Blind Newcastle Mobility Team for helping with distribution of information about the research.

## Author Contributions

**Conceptualization:** Lore Thaler.

**Data curation:** Liam J. Norman, Caitlin Dodsworth, Denise Foresteire, Lore Thaler.

**Formal analysis:** Liam J. Norman, Caitlin Dodsworth, Lore Thaler.

**Funding acquisition:** Lore Thaler.

**Investigation:** Liam J. Norman, Lore Thaler.

**Methodology:** Liam J. Norman, Lore Thaler.

**Project administration:** Liam J. Norman, Lore Thaler.

**Resources:** Liam J. Norman, Lore Thaler.

**Software:** Liam J. Norman, Lore Thaler.

**Supervision:** Liam J. Norman, Lore Thaler.

**Validation:** Liam J. Norman, Lore Thaler.

**Visualization:** Liam J. Norman, Caitlin Dodsworth, Denise Foresteire, Lore Thaler.

**Writing – original draft:** Liam J. Norman, Lore Thaler.

**Writing – review & editing:** Liam J. Norman, Caitlin Dodsworth, Denise Foresteire, Lore Thaler.

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
