## [Decision Letter · Decision Letter 0]

8 Apr 2021

PONE-D-21-05402

Human click-based echolocation: Effects of blindness and age, and real-life implications in a 10-week training program

PLOS ONE

Dear Dr. Thaler,

Thank you for submitting your manuscript to PLOS ONE. After careful consideration, we feel that your paper is acceptable for publication in PLOS ONE. The required corrections are minor and we invite you to take them into consideration in the revised version of the manuscript. 

We look forward to receiving your revised manuscript.

Kind regards,

Maurice Ptito

Academic Editor

PLOS ONE

Journal Requirements:

Reviewers' comments:

Reviewer's Responses to Questions

**Comments to the Author**

1. Is the manuscript technically sound, and do the data support the conclusions?

Reviewer #1: Yes

Reviewer #2: Partly

Reviewer #3: Yes

2. Has the statistical analysis been performed appropriately and rigorously? 

Reviewer #1: Yes

Reviewer #2: Yes

Reviewer #3: No

3. Have the authors made all data underlying the findings in their manuscript fully available?

Reviewer #1: Yes

Reviewer #2: Yes

Reviewer #3: Yes

4. Is the manuscript presented in an intelligible fashion and written in standard English?

Reviewer #1: Yes

Reviewer #2: Yes

Reviewer #3: Yes

5. Review Comments to the Author

Reviewer #1: This research represents an important contribution to the field of vision rehabilitation, as few studies explore the topic of active echolocation in significant depth. The findings (that both blind and sighted participants can improve with training) are especially pertinent given the increased prevalence of late onset vision loss within the population. The manuscript is well written and structured and the methodological decisions well outlined and supported. A few minor comments are provided below.

As a general comment, I am a traditionalist when it comes to the term data (plural) and datum (singular). Please change “data is” or “data has” to “data are” and “data have” throughout the manuscript.

Introduction

Introduction follows a logical progression, beginning with a clear definition of both passive and active echolocation, terms which may be unfamiliar to some readers. The overview of prior research on echolocation including the role of visual deprivation and distinctions between younger and older participants is concise and well written. This provides useful context for the study that follows.

Methods:

For sighted participants: "All reported to have normal or corrected to normal vision and no prior echolocation experience": How were these data collected? Is this self-reported level of vision?

I share the concerns of the previous reviewers regarding the potentially confounding effect of “age” in this analysis. Combined with the stated known findings that older individuals have more difficulty learning and becoming proficient at new and novel skills, this is a significant confounder. However, in reviewing the changes to the manuscript that have been made, it is apparent that the authors have adequately addressed and acknowledged this limitation: The paper now explicitly recognizes that there was a statistically significant difference in age between the two sample groups, and makes it clear that more than 50% of the sighted population having age <= 26. As such, the confounding element of age is expressly illuminated and addressed. Moreover, the authors have demonstrated (in many different ways) that irrespective of the absolute difference in performance between the two groups, significant gains were made by both.

In section 2.2.2.1, it is acknowledged that “for one participant who was sighted (aged 22 yrs) an error occurred and data were not saved.” Given the ultimate finding that binaural differences are a potentially significant factor, the loss of audiometry data for even one participant (given the small sample size) could potentially be a significant issue. Please comment / justify how the loss of this data would (not?) be expected to impact on the results.

Substantial changes were made to the navigation testing paradigm during the last few sessions, including the introduction of “time penalties” for collision errors. In 3.2.4 it is explained that these later sessions were analyzed separately as a result of the changed level of difficulty, but it might be helpful to move that explanation, or at least a foreshadowing that the data for sessions 15-20 on the navigation task were analyzed separately, to section 2.4 where the data analyses are described.

“To validate our paradigms and to benchmark our tasks we also tested seven blind experts in click-based echolocation”: what made an “expert” an “expert” other than that they had been blind for some period of time? How did the researchers identify / recruit / confirm that these participants were experts? Clarification needed here.

Results:

Section 3.1.1 (audiometry results) is not really a “result” of the study, since it is not part of the research question. These “results” are merely demographic descriptors of the participants. Consider moving to section 2.1 where the participants are described.

In Section 3.5, it is noted that 2/12 did not report “improved independence.” Several quotes are then provided from selected participants who did report “improved independence.” For balance, include at least one of the comments from a “no improved independence” respondent, because the comments shed light on the reasons for reporting “no improvement”.

Discussion:

Separate from the issue of age, might there be other reasons why the sighted outperformed the blind? e.g. Were data collected about the orientation and mobility skills of the blind participants prior to training (e.g. spatial orientation, familiarity with the outdoor environment)? Clarification in the methodology and the discussion section would be helpful.

The authors note that the findings of this study highlight the potential benefit of active click-based echolocation training: “Based on our data we therefore suggest that any time is a good time to start learning click-based echolocation, and that it would make good sense to provide training in this skill as part of orientation and mobility instruction to people with vision impairments.” However, it is important to note that click-based echolocation also carries practical limitations related to stigma. The potential limitations of using click-based echolocation should also be briefly acknowledged within this discussion, with reference to any prior research in this domain.

Reviewer #2: There is evidence based on the data presented that sighted individuals can learn echolocation to the same or greater extent than blind individuals. This data could encourage orientation and mobility instructors to incorporate echolocation training to more late-blind individuals.

I appreciate that the authors included the level of functional vision of the participant as that may play a role in their echolocation abilities as well.

However, in my opinion, the evidence is still lacking for the impact that age may have on echolocation skills. The sample size is too small in the groups to adequately conclude that age is not a factor (pg 41, line 912).

The virtual maze data interestingly demonstrates how sighted individuals have the plasticity required to use echolocation skills for simple tasks. However, it has yet to be demonstrated how this applies to realistic scenarios in the everyday life of blind individuals.

pg 9, line 158 A few questions arise regarding the training: How many instructors in echolocation were used and what was their training? If there was more than one instructor, were the participants randomized to different instructors?

p 16 For the natural environment, the extent of the ambient noise present that the participants would typically experience is unknown. Therefore, it is unknown how this could impact their abilities. For instance, would this only be useful in quiet areas, or could it also be used on busy streets?

p 16, lines 327-328 ; Asking the participants to close their eyes is prone to error. Each of the participants should have received blindfolds. Is there a difference between those that used blindfolds and those that did not?

It is unclear as to what constitutes a collision during the natural environment. Blind individuals are typically taught to use their canes to explore their environment, find openings and align themselves. Therefore, any contact that a blind individual had with the cane does not necessarily represent an error, it could represent a curiosity to understand the environment or improve their alignment. If the sighted individuals were told to not contact the wall with their cane during the experiment, they are learning a different technique than is taught by orientation and mobility instructors.

Regarding the experts: what is their level and type of training that they received. What made them experts?

It was interesting to see the perceived impact for the participants in the survey. However, it would have been useful to ask if they are using click-based echolocation in their daily lives.

Overall, I find the results from this study interesting and demonstrate the ability of sighted and late-blind individuals to learn useful echolocation skills. However, there are still many questions for the impact of this training in everyday life and the willingness of participants to apply these skills in indoor and outdoor environments outside of the home.

Reviewer #3: The authors have answered most of my comments well. I am still excited by this manuscript and as in the previous round support its publication.

There are still some issues I am not satisfied with (see below), but the core results are supported and I think my comments can all be readily addressed by the authors.

Main issue:

=========

1. Analysis of the size and orientation tasks, and different distance as qualitatively vs. quantitatively different.

This issue is still open from the previous round.

I understand the authors comments about the ANOVAs, but am not convinced and still feel uneasy about the choice of ANOVAs for analysing the size and orientation tasks due to the different distances.

I acknowledge the authors point about ratio, interval and sphericity, but as a reader I did not think of them thus - my reading, which I presume will be shared by others, was seeing the difference between distances as more categorical in nature - with the task being significantly different at different distances and not just a matter of a linear scale. In my view, when at a different distance the percent correct is a proportion of correctness at a **different task**, and that simply does not fit an ANOVA. A 75% chance at one distance is inherently different the 75% at a different distance! It’s not just a matter of scale.

I think the view of different distances as inherently different rather than intervals in the same tasks also points at several other points where I think the authors misunderstood my previous comments and some of those made by R2.

To give an extreme example, I read sentences like “Furthermore, EEs (n=5) did not perform significantly better than either BCs (U=12; z=-1.901; p=.057) or SCs (U=21; z=-1.310; p=.190) did in the final session.” as saying something like this -

a toddler is as good at shooting hoops as an NBA player since both get 75% of their shots into the basket, despite the toddler using a bucket on the floor as a basket and the NBA player using a regulation hoop. The EEs are clearly qualitatively better due to the difference in the distance!

This is exacerbated by the monotonically rising and success-based transitions of the distance parameter.

Note that the use of non parametric friedman tests does not solve this specific issue, so adding them into the paper will not solve this, and I think it can stay in the Supp or even be omitted for these tasks (my comment on non parametric analysis was more focused on the comparison across groups - and those are now already well incorporated into the manuscript itself).

Thus,

A. I strongly recommend the authors move away from ANOVA based analysis (and trend based analysis) in these two specific tasks. Your points here can be made easily without them and they simply have an uneasy fit to this paradigm.

If they insist not to, they should be prepared to have this argument with future readers and at the very least spotlight this much more explicitly and justify the ANOVAs legitimacy.

B. Authors must correct the phrasing of sentences like in statements like “Furthermore, EEs (n=3) performed significantly better than SCs (U=5; z=-2.042; p=.041) and BCs (U=2; z=-2.311; p=.021) did in the final session.” you simply must clarify that they did so at a totally different distance (and one which does not fit the ANOVA intervals….). Add something like **even though they performed the task at a different and greater distance**.

This is even more important in the following task where you write that “Furthermore, EEs (n=5) did not perform significantly better than either BCs (U=12; z=-1.901; p=.057) or SCs (U=21; z=-1.310; p=.190) did in the final session.” - despite the EE doing a harder task since they were at a different distance, so a statement that they had a similar proportion of correct answers is misleading and kind of meaningless.

I again emphasize that despite my issues with the analysis, looking at the results show that the conclusions the authors draw from them are obviously correct. My issue is with their methods, but the actual results is clearly true.

Minor issues:

(note that this may look long, but most of the bullet points below are more in the nature of recommendations and suggestions then critical fixes)

2. Please clarify in the phrasing directly that the EE group was not part of the ANOVAs. You fixed this in terms of analysis, but the text describing the analysis is still not clear enough on this point.

3. Please avoid treating numbers right around 0.05 as inherently different - e.g. "did not perform significantly better than either BCs (U=12; z=-1.901; p=.057) clearly trends to significance, while “ but SCs were significantly faster than EEs (U=9; z=- 539 2.018; p=.044)” is only borderline significant. While this issue is common in the literature, I would recommend rephrasing accordingly.

4. Please add the method of multiple comparisons corrections which you used in your stats

5. I would prefer if the text had absolute values rather than just effect sizes and stats. These can indeed be seen in the figures, but given that your effects are behaviorally clearly different and not just statistically so - e.g. participants improved from 55% to 85% correct has a much stronger message then saying that subjects significantly improved with a nice p-value.

Note that you already do this well in some places, such as " Throughout these sessions, SCs had fewer collisions (mean: 3.853; SD: 3.18) than BCs (mean: 6.951; SD: 3.18; F(1,24)=6.138; p=.021, η2 576 p: .204)", but it is missing in most of them. Please add this in.

6. In my opinion section 3.5 is by far the most important result, but still feels a bit drowned out by the rest of the stats. This is your paper and not mine, but I would recommend highlighting this result much more. I cannot think of other work with blind users which got such enthusiastic feedback

(and if you have a way to contact these subjects again now with more questions, including some more quantitative assessments, and the results still hold that should also be published in the future)

7. "suggesting that differences in binaural hearing might be a limiting factor for participants learning echolocation skills through computer-based training" - this is a critical result, it suggests a standard screening procedure before starting training - do you see this effect also in experts? previous studies? I would highlight the potential for a future screening test based on this and the need to further test this issue specifically in the future.

8. If possible, please attribute the quotes to the participant numbers from the table (or at least clarify that each bullet is from a different subject).

9. Reading through the discussion between the authors and R2, I would like to make 3 quick comments:

a. Regarding the participant quotes - I agree with the authors, the quotes should be kept in and not summarized as R2 suggests. These quotes are very powerful (see my comment above), and summarizing them would detract from the effect.

b. Diagram of the tasks - R2 suggests a diagram of the general training progרam. The authors answer that “due to the heterogeneity of the order in which tasks were conducted a systematic figure will not be useful”. I respectfully disagree with the authors - I feel that providing a figure of the general outline, with a series of what a single participant experienced (and with a clear comment/graphics in the figure stating that is only a representative example, and that the order was randomized across subjects) would add clarity. It would also let the authors clearly introduce the post 3 month survey earlier as part of this figure.

c. " Importantly, every blind control participant improved with training." - this is stated in the answer to r2, but not in the paper, and it's an important result which might be worth adding in briefly if you can.

6. PLOS authors have the option to publish the peer review history of their article (what does this mean?). If published, this will include your full peer review and any attached files.

Reviewer #1: No

Reviewer #2: **Yes: **Joseph Paul Nemargut

Reviewer #3: No

---

## [Author Response · Author response to Decision Letter 0]

15 Apr 2021

we have also submitted a word document with the response to reviews. that document contains formatting which makes it easier to read our responses (they are in red font, whilst reviewer comments are in black font).

Below you will find our responses in red to each comment made by the reviewers.

Reviewer #1: This research represents an important contribution to the field of vision rehabilitation, as few studies explore the topic of active echolocation in significant depth. The findings (that both blind and sighted participants can improve with training) are especially pertinent given the increased prevalence of late onset vision loss within the population. The manuscript is well written and structured and the methodological decisions well outlined and supported. A few minor comments are provided below.

As a general comment, I am a traditionalist when it comes to the term data (plural) and datum (singular). Please change “data is” or “data has” to “data are” and “data have” throughout the manuscript.

We have now addressed this in all instances and use plural throughout the manuscript.

Introduction

Introduction follows a logical progression, beginning with a clear definition of both passive and active echolocation, terms which may be unfamiliar to some readers. The overview of prior research on echolocation including the role of visual deprivation and distinctions between younger and older participants is concise and well written. This provides useful context for the study that follows.

Thank you for the positive feedback.

Methods:

For sighted participants: "All reported to have normal or corrected to normal vision and no prior echolocation experience": How were these data collected? Is this self-reported level of vision?

We have now clarified that this was based on self-report:

Page 6, line 118

“All reported to have normal or corrected to normal vision and no prior echolocation experience (based on self-report).”

I share the concerns of the previous reviewers regarding the potentially confounding effect of “age” in this analysis. Combined with the stated known findings that older individuals have more difficulty learning and becoming proficient at new and novel skills, this is a significant confounder. However, in reviewing the changes to the manuscript that have been made, it is apparent that the authors have adequately addressed and acknowledged this limitation: The paper now explicitly recognizes that there was a statistically significant difference in age between the two sample groups, and makes it clear that more than 50% of the sighted population having age <= 26. As such, the confounding element of age is expressly illuminated and addressed. Moreover, the authors have demonstrated (in many different ways) that irrespective of the absolute difference in performance between the two groups, significant gains were made by both.

Thank you

In section 2.2.2.1, it is acknowledged that “for one participant who was sighted (aged 22 yrs) an error occurred and data were not saved.” Given the ultimate finding that binaural differences are a potentially significant factor, the loss of audiometry data for even one participant (given the small sample size) could potentially be a significant issue. Please comment / justify how the loss of this data would (not?) be expected to impact on the results.

When audiometry data were measured and visually inspected at the end of the audiometric measurement, there was no reason to suspect hearing loss either monaural or binaural for this participant. Yet, as stated in the manuscript, due to an error data were not saved for numerical analysis. The age of this sighted participant (22 yrs) places them into an age range for which we have multiple participants in the sighted group (i.e. 21yrs � n=2, 22yrs � n=2 (incl. this participant), 23yrs � n=1). Binaural hearing was correlated with age. Furthermore, on other aspects of the study this participant performed in line with what these other participants did. Thus, apart from losing a data point which weakens statistical power, we do not expect this to impact on our results. 

Substantial changes were made to the navigation testing paradigm during the last few sessions, including the introduction of “time penalties” for collision errors. In 3.2.4 it is explained that these later sessions were analyzed separately as a result of the changed level of difficulty, but it might be helpful to move that explanation, or at least a foreshadowing that the data for sessions 15-20 on the navigation task were analyzed separately, to section 2.4 where the data analyses are described.

Yes, thank you for pointing this out. We have added: 

Page 21, line 423

“For the virtual navigation task, due to the differences in task structure introduced from session 15 onwards, data were analysed separately for sessions 1-14 and for 15-20.”

“To validate our paradigms and to benchmark our tasks we also tested seven blind experts in click-based echolocation”: what made an “expert” an “expert” other than that they had been blind for some period of time? How did the researchers identify / recruit / confirm that these participants were experts? Clarification needed here.

Added: 

Page 6, line 133

“Our requirements for classing an individual as an echolocation expert were that they reported using click-based echolocation on a daily basis for more than 10 years.” 

Results:

Section 3.1.1 (audiometry results) is not really a “result” of the study, since it is not part of the research question. These “results” are merely demographic descriptors of the participants. Consider moving to section 2.1 where the participants are described.

Following the reviewers suggestion we have now moved the “results” sections on audiometry and DFM/DCI tests to 2.2.1 to accompany the methodological descriptions of how these data were acquired. 

In Section 3.5, it is noted that 2/12 did not report “improved independence.” Several quotes are then provided from selected participants who did report “improved independence.” For balance, include at least one of the comments from a “no improved independence” respondent, because the comments shed light on the reasons for reporting “no improvement”.

Yes, thank you for pointing this out! We have added comments to this effect. 

Page 41, line 925

e.g. “BC5: The reason why I felt unable to answer yes to the above questions is due to the fact that having worked with several mobility and rehabilitation instructors over the years, I felt confident that all the methods and techniques I have learned over time work well enough to help me feel confident to get out and about safely and independently, even before I took part in the research.”

Discussion:

Separate from the issue of age, might there be other reasons why the sighted outperformed the blind? e.g. Were data collected about the orientation and mobility skills of the blind participants prior to training (e.g. spatial orientation, familiarity with the outdoor environment)? Clarification in the methodology and the discussion section would be helpful.

All our BCs were independent travellers and all had received mobility and orientation training as part of visual impairment habilitation and rehabilitation that is provided to people who are visually impaired in the UK. We have added this information to the section describing our participants.

Page 6, line 125

“All our blind participants were independent travellers and all had received mobility and orientation training as part of visual impairment (VI) habilitation and VI rehabilitation that is provided to people with VI in the UK.”

We did not take specific measures related to these factors, however. We have now made this clear to readers in the paper, specifically including a paragraph in the discussion section on the possible role that other factors might have played:

Page 45, line 1028

“There are other possible reasons that might explain performance differences between our BCs and SCs beyond blindness per se that we did not address in this study. For example, even though all our BCs were independent travellers, it is still possible that our SCs enjoyed greater levels of independent mobility compared to our BCs. In fact, our survey results to a degree attest to this in particular before taking part in the research, since the majority BCs point out that before taking part they relied more on other people in unfamiliar or novel situations. This in turn might manifest as a group difference in how people approach a novel training situation, which in turn may affect learning. We did not take measures relating to such factors at the beginning of our training, thus we cannot establish or rule out their effects, but it is nonetheless possible that SCs might have an advantage in our study for reasons beyond blindness and/or age. Although it is difficult to identify the specific reasons for any group differences, what is clear in our results is that both sighted and blind people can improve considerably in their echolocation ability with training.”

The authors note that the findings of this study highlight the potential benefit of active click-based echolocation training: “Based on our data we therefore suggest that any time is a good time to start learning click-based echolocation, and that it would make good sense to provide training in this skill as part of orientation and mobility instruction to people with vision impairments.” However, it is important to note that click-based echolocation also carries practical limitations related to stigma. The potential limitations of using click-based echolocation should also be briefly acknowledged within this discussion, with reference to any prior research in this domain.

We thank the reviewer for this comment. 

We have added a discussion on this point to the final paragraph:

Page 48, line 1106

“Echolocation is currently not taught as part of mobility training and rehabilitation for blind people, and there is the possibility that some people are reluctant to use click-based echolocation due to a perceived stigma around making the clicks in social environments. Yet, blind people who use echolocation do so in a way that is adaptive to the social situation [17] and, as shown by our survey results, people new to echolocation are confident to use it in situations with other people. The potential barriers relating to perceived stigma are, therefore, perhaps much smaller than previously thought.”

Reviewer #2: There is evidence based on the data presented that sighted individuals can learn echolocation to the same or greater extent than blind individuals. This data could encourage orientation and mobility instructors to incorporate echolocation training to more late-blind individuals.

I appreciate that the authors included the level of functional vision of the participant as that may play a role in their echolocation abilities as well.

However, in my opinion, the evidence is still lacking for the impact that age may have on echolocation skills. The sample size is too small in the groups to adequately conclude that age is not a factor (pg 41, line 912).

We agree that our sample size was small. Yet, despite the small sample we did find effects of age in aspects of our study e.g. computer based task, completion times in particular. Thus, we do not think that the sample size per se is a limiting factor, but that there are definite differences in the tasks and how this is affected by age. In response to the reviewer’s concern we have added the following caveat to this section in the discussion at this paragraph, which now reads:

Page 45, line 1010

“Importantly, when we quantified to degree to which participants improved from session 1 to session 20 in their abilities across each of the tasks, there was no evidence for an association between age and performance in the practical tasks, suggesting that age is unlikely to be a limiting factor in learning these echolocation abilities. We did find, however, that younger age was associated with greater improvements in the time it took people to navigate virtual mazes. Thus, even though our sample was small, the tasks and measures we used showed differences in their sensitivity to age, with active tasks showing no effects and computer-based tasks showing age effects with respect to completion times. Given our small sample, however, further research with larger samples and thus greater statistical power should continue investigating the role played by age for learning a novel sensory skill like echolocation.”

The virtual maze data interestingly demonstrates how sighted individuals have the plasticity required to use echolocation skills for simple tasks. However, it has yet to be demonstrated how this applies to realistic scenarios in the everyday life of blind individuals.

Yes, we agree with the reviewer. 

Whilst our survey results demonstrate that the training as a whole (i.e. including both computer based and practical tasks) affects realistic scenarios in the everyday life of blind individuals, we have not yet investigated if and how training of only the computer based virtual navigation task would transfer into real life. Future research should investigate these issues further. 

We have now added to the discussion to address this:

Page 48, line 1089

“It is also important to quantify the degree to which abilities acquired in computer-based tasks translate into real-world settings. The results from our virtual navigation task establish that performance generalises very well from untrained to novel environments regardless of the person’s sightedness, but we have not quantified how well these skills transfer to realistic scenarios in the physical world. Whilst our survey results demonstrate that the training as a whole (i.e. including both computer based and practical tasks) affects realistic scenarios in the everyday life of blind individuals, we did not investigate if and how training of only the computer based virtual navigation task would transfer into real life. This should be an important avenue for subsequent research.”

pg 9, line 158 A few questions arise regarding the training: How many instructors in echolocation were used and what was their training? If there was more than one instructor, were the participants randomized to different instructors?

We have added the following to the methods section:

Page 10, line 175

“Three of the authors (LN, LT and CD) led instruction. To ensure consistency, the principal investigator (LT) gave in house training to all instructors prior to any training commencing, and (determined by availability of the instructors) at least two instructors were present for any session. Instructors were not assigned in any specific fashion to participants or sessions, but this was determined by availability.”

p 16 For the natural environment, the extent of the ambient noise present that the participants would typically experience is unknown. Therefore, it is unknown how this could impact their abilities. For instance, would this only be useful in quiet areas, or could it also be used on busy streets?

We have added the following to the methods section:

Page 19, line 396

“The ambient noise present in these indoor and outdoor environments was not recorded, but all participants experienced a range of light to moderate to loud ambient sound levels in the environments that they sampled. Specifically, the environments ranged from empty to moderately busy to very busy corridors, classrooms, garden areas, communal areas, campuses, and outdoor pedestrian areas. Consistent with our observation that participants coped with these varied levels of ambient noise, we have also shown in laboratory based work [40] that intensity of click emissions can compensate for the presence of background noise in human echolocation.”

p 16, lines 327-328 ; Asking the participants to close their eyes is prone to error. Each of the participants should have received blindfolds. Is there a difference between those that used blindfolds and those that did not?

Yes, we agree and normally we would have asked everyone to wear a blindfold for the outdoor navigation task just as we had asked for all of the lab based tasks (where everyone wore blindfolds), but some blind participants were apprehensive about wearing a blindfold whilst navigating in an unfamiliar public space. Thus, we gave them the option to close their eyes instead of wearing a blindfold.

We have clarified this statement in the revised version:

Page 18, line 376

“To avoid apprehensiveness about wearing a blindfold whilst navigating in an unfamiliar public space, participants could choose to either wear a blindfold or close their eyes for this part of the training. All participants with normal vision and nine participants who were blind wore a blindfold. The remaining three (BC5, BC6, BC9) closed their eyes.”

It is unclear as to what constitutes a collision during the natural environment. Blind individuals are typically taught to use their canes to explore their environment, find openings and align themselves. Therefore, any contact that a blind individual had with the cane does not necessarily represent an error, it could represent a curiosity to understand the environment or improve their alignment. If the sighted individuals were told to not contact the wall with their cane during the experiment, they are learning a different technique than is taught by orientation and mobility instructors.

In the natural environment all participants were encouraged to explore the environment in any way they wanted, including the long cane. As the reviewer points out, this is a normal part of active exploration. Only in the virtual environment, i.e. the computer based task, did we use collisions as a measure of performance, and in the virtual task there was no long cane, i.e. everything was based on sound. This was explicitly told to all participants. 

We have added this to the manuscript text on page 19, lines 382:

“All participants were encouraged to explore the environment in any way they wanted, including the long cane. ” 

Regarding the experts: what is their level and type of training that they received. What made them experts?

We have added the following clarification to the methods section: 

Page 6, line 131

“Our requirements for classing an individual as an echolocation expert were that they reported using click-based echolocation on a daily basis for more than 10 years.” 

It was interesting to see the perceived impact for the participants in the survey. However, it would have been useful to ask if they are using click-based echolocation in their daily lives.

In most instances the survey responses themselves document people’s use of the clicks and echoes in their daily lives, e.g. 

Page 38, line 849

“I use the clicks and echoes to find doors and entrances, to determine what is around; I use it more in quite environments, it is less useful for me in noisy places.”

In some places, however, participants answers refer to the use of click-based echolocation as ‘it’, because it was understood from the question and context that this is about click-based echolocation, e.g.

Page 38, line 837

“Helps with spatial awareness; when I am out it is useful for example for finding landmarks and side streets.”

To clarify, we have indicated (in parenthesis) in the survey responses where this is the case to improve clarity. We have indicated this also in the supplemental data file.

“Helps with spatial awareness; when I am out it (i.e. click-based echolocation) is useful for example for finding landmarks and side streets.”

Overall, I find the results from this study interesting and demonstrate the ability of sighted and late-blind individuals to learn useful echolocation skills. However, there are still many questions for the impact of this training in everyday life and the willingness of participants to apply these skills in indoor and outdoor environments outside of the home.

Thank you – we appreciate your comments very much. 

The survey responses clearly demonstrate that participants apply the skills in indoor and outdoor environments outside of their home. 

To improve clarity on those survey responses where participants referred to click-based echolocation as ‘it’ we have added ‘click-based echolocation’ in parenthesis. We hope that this addresses the reviewer’s concern.

Reviewer #3: The authors have answered most of my comments well. I am still excited by this manuscript and as in the previous round support its publication.

There are still some issues I am not satisfied with (see below), but the core results are supported and I think my comments can all be readily addressed by the authors.

Main issue:

=========

1. Analysis of the size and orientation tasks, and different distance as qualitatively vs. quantitatively different.

This issue is still open from the previous round.

I understand the authors comments about the ANOVAs, but am not convinced and still feel uneasy about the choice of ANOVAs for analysing the size and orientation tasks due to the different distances.

I acknowledge the authors point about ratio, interval and sphericity, but as a reader I did not think of them thus - my reading, which I presume will be shared by others, was seeing the difference between distances as more categorical in nature - with the task being significantly different at different distances and not just a matter of a linear scale. In my view, when at a different distance the percent correct is a proportion of correctness at a **different task**, and that simply does not fit an ANOVA. A 75% chance at one distance is inherently different the 75% at a different distance! It’s not just a matter of scale.

I think the view of different distances as inherently different rather than intervals in the same tasks also points at several other points where I think the authors misunderstood my previous comments and some of those made by R2.

To give an extreme example, I read sentences like “Furthermore, EEs (n=5) did not perform significantly better than either BCs (U=12; z=-1.901; p=.057) or SCs (U=21; z=-1.310; p=.190) did in the final session.” as saying something like this -

a toddler is as good at shooting hoops as an NBA player since both get 75% of their shots into the basket, despite the toddler using a bucket on the floor as a basket and the NBA player using a regulation hoop. The EEs are clearly qualitatively better due to the difference in the distance!

This is exacerbated by the monotonically rising and success-based transitions of the distance parameter.

Note that the use of non parametric friedman tests does not solve this specific issue, so adding them into the paper will not solve this, and I think it can stay in the Supp or even be omitted for these tasks (my comment on non parametric analysis was more focused on the comparison across groups - and those are now already well incorporated into the manuscript itself).

Thus,

A. I strongly recommend the authors move away from ANOVA based analysis (and trend based analysis) in these two specific tasks. Your points here can be made easily without them and they simply have an uneasy fit to this paradigm.

If they insist not to, they should be prepared to have this argument with future readers and at the very least spotlight this much more explicitly and justify the ANOVAs legitimacy.

We now understand the concern raised. We had indeed misunderstood in the last round of reviews.

We do not think that the change in distance affects the nature of the task in the fundamental way that is the concern to the reviewer. Instead, we feel that increasing the distance makes it more difficult whilst keeping the essence of the task the same. The task becomes more difficult by virtue of the reduction in echo intensity – and we and others have shown in previous experiments that echo detection is poorer at greater distances (e.g. Norman & Thaler, 2018; Schenkman & Nilsson, 2010). This assessment is also based on our subjective judgment of these tasks, which we have tested extensively when developing these paradigms. 

Ref:

Norman, L. & Thaler, L. (2018). Human echolocation for target detection is more accurate with emissions containing higher spectral frequencies, and this is explained by echo intensity. i-Perception 9(3): 1-18.

Schenkman, B.N., Nilsson, M.E. (2010) Human echolocation: Blind and sighted persons' ability to detect sounds recorded in the presence of a reflecting object. Perception, 39: 483-501.

Our main result is that accuracy increases with training. We show this in all tasks, despite the fact that for some subjects the distance increases (and therefore difficulty also increases). Thus, the ANOVA analysis we chose is a conservative test, because it ignores this increase in difficulty. We do, acknowledge the inherent complexity in comparing accuracy at one distance to that at another, however, and the particular caveat this presents when comparing echolocation experts’ performance to that of non-experts. For this reason, we have now included clear statements to stress that in these cases (as suggested by the reviewer):

e.g. Page 25, line 532

 “EEs also did not perform better than SCs did in the final session (U=21; z=-1.310; p=.190). Note also that EEs did performed the task at a different and greater distance (i.e. at 100 cm) than the BCs and SCs did.”

Finally, there are also practical issues to consider around which statistical analyses would be appropriate if we were to treat the task as ‘different’ at different distances. Specifically, we do not see which inferential statistical analysis we could possibly use if we assume that a change in distance causes a fundamental change in the task across sessions, because this would make it impossible to compare across sessions at all. We would then be left with the option to only provide descriptive statistics of the data, which we already include in the figures and (now) within the main text.

Thus, what we have done is to follow the reviewer’s suggestion and to spotlight the issue and justify the ANOVAs legitimacy for analysis of the proportion correct data for these tasks. We now write in the results section for proportion correct data:

Page 23, line 484

“In our paradigm, distance could only increase (and not decrease) depending on performance, i.e. the task could only get more difficult. Yet, our analysis of proportion correct data with ANOVA disregards changes in distance, which is equivalent to assuming that the task stays equally ‘easy’ at all times. It follows that our ANOVA analysis gives a conservative estimate of participants’ improvement, as it assumes that the task does not increase in difficulty with target distance.”

Page 25, line 534

“Again, just as for the size discrimination task, our analysis of proportion correct data with ANOVA disregards changes in distance, which is equivalent to assuming that the task stays equally ‘easy’ at all times, and it follows that our ANOVA analysis gives a conservative estimate of participants’ improvement.” 

B. Authors must correct the phrasing of sentences like in statements like “Furthermore, EEs (n=3) performed significantly better than SCs (U=5; z=-2.042; p=.041) and BCs (U=2; z=-2.311; p=.021) did in the final session.” you simply must clarify that they did so at a totally different distance (and one which does not fit the ANOVA intervals….). Add something like **even though they performed the task at a different and greater distance**.

This is even more important in the following task where you write that “Furthermore, EEs (n=5) did not perform significantly better than either BCs (U=12; z=-1.901; p=.057) or SCs (U=21; z=-1.310; p=.190) did in the final session.” - despite the EE doing a harder task since they were at a different distance, so a statement that they had a similar proportion of correct answers is misleading and kind of meaningless.

As the reviewer has suggested, we have added the clarifying statements to emphasise that comparisons between experts and non-experts were made at different distances.

Page 23, line 481

“Furthermore, EEs (n=3) performed significantly better than SCs (U=5; z=-2.042; p=.041) and BCs (U=2; z=-2.311; p=.021) did in the final session, even though they performed the task at a different and greater distance (i.e. at 100 cm).” 

Page 25, line 530

“Furthermore, although EEs (n=5) did not perform significantly better than BCs did in the final session (U=12; z=-1.901; p=.057), note that this effect is only marginally non-significant. EEs also did not perform better than SCs did in the final session (U=21; z=-1.310; p=.190). Note also that EEs performed the task at a different and greater distance (i.e. at 100 cm) than the BCs and SCs did.”

I again emphasize that despite my issues with the analysis, looking at the results show that the conclusions the authors draw from them are obviously correct. My issue is with their methods, but the actual results is clearly true.

Thank you again for raising this comment. We hope that our current revision now addresses this appropriately. 

Minor issues:

(note that this may look long, but most of the bullet points below are more in the nature of recommendations and suggestions then critical fixes)

2. Please clarify in the phrasing directly that the EE group was not part of the ANOVAs. You fixed this in terms of analysis, but the text describing the analysis is still not clear enough on this point.

We have added this clarification to the Data Analysis section:

Page 21, line 416

“Our main statistical analyses across training tasks were conducted using mixed model ANOVA with ‘session’ as a within subject factor and ‘group’ as a between subject factor, which were conducted with SPSS v26 (EEs were not included in these ANOVAs).”

And to the individual sections reporting each analysis:

Page 23, line 475

“Based on the overall ANOVA analysis (not including EE group data), there was no difference between participant groups (F(1,24)=2.801; p=.107, η2p: .105) or interaction between group and session (FGG (6.831, 163.955)=1.422; p=.201; η2p: .056).”

3. Please avoid treating numbers right around 0.05 as inherently different - e.g. "did not perform significantly better than either BCs (U=12; z=-1.901; p=.057) clearly trends to significance, while “ but SCs were significantly faster than EEs (U=9; z=- 539 2.018; p=.044)” is only borderline significant. While this issue is common in the literature, I would recommend rephrasing accordingly.

We have modified how we report such marginal results:

Page 25, line 530:

“Furthermore, although EEs (n=5) did not perform significantly better than BCs did in the final session (U=12; z=-1.901; p=.057), note that this effect is only marginally non-significant. EEs also did not perform better than SCs did in the final session (U=21; z=-1.310; p=.190). Note also that EEs performed the task at a different and greater distance (i.e. at 100 cm) than the BCs and SCs did.” 

Page 28, line 592

“Furthermore, BCs did not significantly differ in completion time compared to EEs (n=4; U=17; z=-.849; p=.396), but SCs were significantly faster than EEs (U=9; z=-2.018; p=.044), although this effect was only marginally significant.”

Page 29, line 618

“With age as covariate, the effect of group became non-significant (F(1,23)=3.687; p=.067; η2p: .138), although only marginally so, and the effect of age itself was significant (F(1,23)=7.340; p=.013; η2p: .242), and positive (average standardized beta weight for the covariate: 2.51) suggesting that older age is associated with longer maze completion times, and that this might be responsible for differences in performance between SCs and BCs in this task.”

Page 31, line 678

“Focusing only on performance in session 14, there was no significant difference between BCs and SCs (t(12.839)=2.207; p=.064), although this was only marginally non-significant.”

4. Please add the method of multiple comparisons corrections which you used in your stats

We used Bonferroni correction.

We have added this:

Page 21, line 425:

“For analyses involving multiple comparisons, Bonferroni correction was applied.”

5. I would prefer if the text had absolute values rather than just effect sizes and stats. These can indeed be seen in the figures, but given that your effects are behaviorally clearly different and not just statistically so - e.g. participants improved from 55% to 85% correct has a much stronger message then saying that subjects significantly improved with a nice p-value.

Note that you already do this well in some places, such as " Throughout these sessions, SCs had fewer collisions (mean: 3.853; SD: 3.18) than BCs (mean: 6.951; SD: 3.18; F(1,24)=6.138; p=.021, η2 576 p: .204)", but it is missing in most of them. Please add this in.

We have added these absolute values to the parts in the text where we report significant effects for accuracy in the two practical tasks, and where we report significant effects for completion time, number of collisions, and prop correct in the virtual maze task:

e.g. page 23, line 473; Size task accuracy

“On average, proportion correct improved from .54 (session 1) to .79 (session 20).”

6. In my opinion section 3.5 is by far the most important result, but still feels a bit drowned out by the rest of the stats. This is your paper and not mine, but I would recommend highlighting this result much more. I cannot think of other work with blind users which got such enthusiastic feedback

(and if you have a way to contact these subjects again now with more questions, including some more quantitative assessments, and the results still hold that should also be published in the future)

Thank you so much for the positive comments! 

To highlight this aspect of our study more, we have moved the sentence about the survey to an earlier position in the abstract, added a short statement at the beginning of section 3.5 and one at the beginning of the discussion to highlight the importance of these data:

Page 2, line 26:

“Blind participants also took part in a 3-month follow up survey assessing the effects of the training on their daily life.”

Page 37, line 821

“An important consideration is whether learning these echolocation skills are associated with real-life improvements in people’s mobility, independence, and/or well-being. Thus we here took the approach to follow up with participants who are blind 3 months after having taken part in the research and to ask them if and how the training in click-based echolocation affected their mobility, independence and wellbeing in their daily lives. All 12 blind participants took part in our follow up survey. All answers and responses are provided in Supplementary Material S3.”

Page 43, line 978

“Overall, both sighted and blind participants showed clear improvements in echolocation ability across a range of practical and virtual tasks in a 10-week training program. Most importantly, our blind participants also reported in a follow-up survey that learning these skills made a positive impact on their mobility, independence and well-being.”

7. "suggesting that differences in binaural hearing might be a limiting factor for participants learning echolocation skills through computer-based training" - this is a critical result, it suggests a standard screening procedure before starting training - do you see this effect also in experts? previous studies? I would highlight the potential for a future screening test based on this and the need to further test this issue specifically in the future.

We do not have any data relating to our experts, and - apart from the one study already cited - we are also not aware of any previous studies that report the same finding. The effect is limited to the computer based tasks, i.e. there was no relationship for the practical tasks. Furthermore, based on survey responses all participants who were blind benefitted, i.e. there did not seem to be an impact on daily life benefits as far as we can tell based on the responses participants gave. 

We have added a statement to the discussion to stress the potential significance of this finding, but to also emphasize that the effect does not seem relevant for active echolocation and real-life implications:

Page 45, line 1025

“Future research should aim to replicate this result in a larger sample. Importantly, the effect we found did not play a role for active echolocation and real-life implications assessed through a survey, suggesting that any future research needs to assess these separate aspects carefully.”

8. If possible, please attribute the quotes to the participant numbers from the table (or at least clarify that each bullet is from a different subject).

We have added this information. 

9. Reading through the discussion between the authors and R2, I would like to make 3 quick comments:

a. Regarding the participant quotes - I agree with the authors, the quotes should be kept in and not summarized as R2 suggests. These quotes are very powerful (see my comment above), and summarizing them would detract from the effect.

Thank you again for the positive comments relating to the significance of the survey data.

b. Diagram of the tasks - R2 suggests a diagram of the general training progרam. The authors answer that “due to the heterogeneity of the order in which tasks were conducted a systematic figure will not be useful”. I respectfully disagree with the authors - I feel that providing a figure of the general outline, with a series of what a single participant experienced (and with a clear comment/graphics in the figure stating that is only a representative example, and that the order was randomized across subjects) would add clarity. It would also let the authors clearly introduce the post 3 month survey earlier as part of this figure.

We have now included a figure (now figure 1 in the revised version) to illustrate the general procedure for data collection for a typical participant.

c. " Importantly, every blind control participant improved with training." - this is stated in the answer to r2, but not in the paper, and it's an important result which might be worth adding in briefly if you can.

We have now added a sentence in the beginning of the general discussion to speak to this:

Page 43, line 978

“In fact, every blind control participant improved with training.”

---

## [Editor Report · Decision Letter 1]

14 May 2021

Human click-based echolocation: Effects of blindness and age, and real-life implications in a 10-week training program

PONE-D-21-05402R1

Dear Dr. Thaler,

We’re pleased to inform you that your manuscript has been judged scientifically suitable for publication and will be formally accepted for publication once it meets all outstanding technical requirements.

Kind regards,

Maurice Ptito

Academic Editor

PLOS ONE
---

## [Editor Report · Acceptance letter]

18 May 2021

PONE-D-21-05402R1 

Human click-based echolocation: Effects of blindness and age, and real-life implications in a 10-week training program 

Dear Dr. Thaler:

I'm pleased to inform you that your manuscript has been deemed suitable for publication in PLOS ONE. Congratulations! Your manuscript is now with our production department. 

Kind regards, 

on behalf of

Dr. Maurice Ptito 

Academic Editor

PLOS ONE